# Structure of phospholipase Cε reveals an integrated RA1 domain and previously unidentified regulatory elements

Ngango Y. Rugema[1,3], Elisabeth E. Garland-Kuntz[1,3], Monita Sieng[1], Kaushik Muralidharan[2], Michelle M. Van Camp[1], Hannah O'Neill[1], William Mbongo[1], Arielle F. Selvia[1], Andrea T. Marti[2], Amanda Everly[1], Emmanda McKenzie[1] & Angeline M. Lyon [1,2 ✉]

Phospholipase Cε (PLCε) generates lipid-derived second messengers at the plasma and perinuclear membranes in the cardiovascular system. It is activated in response to a wide variety of signals, such as those conveyed by Rap1A and Ras, through a mechanism that involves its C-terminal Ras association (RA) domains (RA1 and RA2). However, the complexity and size of PLCε has hindered its structural and functional analysis. Herein, we report the 2.7 Å crystal structure of the minimal fragment of PLCε that retains basal activity. This structure includes the RA1 domain, which forms extensive interactions with other core domains. A conserved amphipathic helix in the autoregulatory X–Y linker of PLCε is also revealed, which we show modulates activity in vitro and in cells. The studies provide the structural framework for the core of this critical cardiovascular enzyme that will allow for a better understanding of its regulation and roles in disease.

[1] Department of Chemistry, Purdue University, West Lafayette 47907 IN, USA. [2] Department of Biological Sciences, Purdue University, West Lafayette 47907 IN, USA. [3] These authors contributed equally: Ngango Y. Rugema, Elisabeth E. Garland-Kuntz. ✉email: lyonam@purdue.edu

The phospholipase C (PLC) superfamily hydrolyzes phosphatidylinositol (PI) lipids to produce second messengers that increase intracellular $Ca^{2+}$ and activate protein kinase C (PKC)[1,2]. Many PLC enzymes are activated in response to extracellular stimuli conveyed by G-protein-coupled receptors (GPCRs) or receptor tyrosine kinases (RTKs), and thus contribute to numerous processes, including cell proliferation, differentiation, and survival[1–4]. PLCε is unusual in that it can integrate signals from both GPCRs and RTKs. The enzyme has also emerged as an important regulator of cardiovascular function because it is required for maximum cardiac contractility, and changes in its expression or activation result in cardiac hypertrophy and heart failure[5–8]. Polymorphisms within PLCε are also linked to an increased risk of gastric and esophageal cancers, potentially mediated by changes in its basal activity and/or sensitivity to Ras-dependent activation[9–11].

PLCε shares four core domains common to most PLCs, including a pleckstrin homology (PH) domain, followed by four tandem EF hand repeats (EF1-4), the catalytic TIM barrel domain (split by an autoregulatory "X–Y linker"), and a C2 domain (Fig. 1a)[1,2,12]. In PLCε, these core domains are flanked by regions that confer responsiveness to different signal transduction pathways. The N-terminal region contains a CDC25 domain that acts as a guanine nucleotide exchange factor (GEF) for the Rap1A GTPase. This domain is essential for sustained PI hydrolysis at the perinuclear and Golgi membranes in cardiomyocytes[13–17]. The C-terminal region contains two Ras association (RA) domains (RA1 and RA2) that have been proposed to autoinhibit basal activity[18], interact with muscle-specific A-kinase anchoring protein (mAKAP) at the perinuclear membrane[19], and bind activated Rap1A and Ras[13,20,21].

Structural insights into PLCε, and the molecular mechanisms regulating its basal and G protein-stimulated activity have remained poorly understood relative to other PLCs. This is due to its size (220 kDa), its complex structure (eight predicted domains), ill-defined domain boundaries, and the small amount of active material that could be isolated for biophysical studies. Thus, early work focused on characterizing the structure and functions of individual domains. NMR structures of each RA domain and a crystal structure of activated H-Ras bound to the RA2 domain are the only high-resolution insights into the enzyme[18]. However, these structures are limited in the information they can provide with respect to the mechanisms by which these domains contribute to regulation of lipase basal activity, membrane association, and/or activation by G proteins. We previously used small-angle X-ray scattering (SAXS) and negative stain electron microscopy (EM) to investigate the solution architecture and conformational dynamics of various catalytically active fragments of PLCε[22]. However, the resolution limit of these techniques (20–40 Å) is insufficient to position individual domains or to define the specific regulatory interactions between domains.

In this work, we first used a domain deletion approach to demonstrate that the RA1 and RA2 domains play distinct roles in modulating PLCε basal activity and contributing to its overall stability. These studies led to the identification of a truncation variant of PLCε retaining the RA1 domain that expressed at high levels and was amenable to high-resolution crystal structure determination. The resulting model reveals that a C2-RA1 linker region and the RA1 domain make extensive interactions with EF hands 3/4 (EF3/4), the C2 domain, and the TIM barrel that are important for maintaining the structural and functional integrity of the protein. Thus, in PLCε, the RA1 domain is an integral part of the catalytic core. In addition, the PLCε autoinhibitory X–Y linker contains a conserved amphipathic helix that can form protein–protein interactions and modulate basal activity in vitro

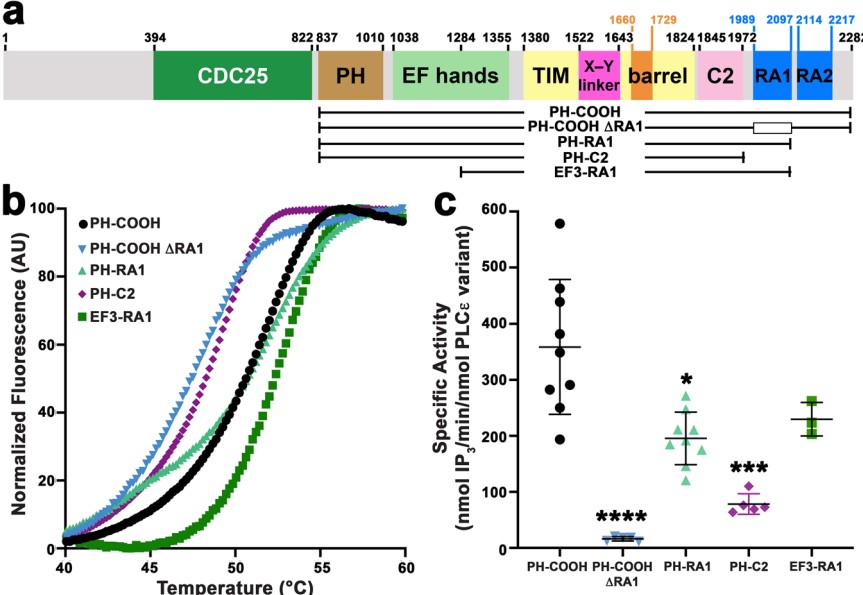

**Fig. 1 The PLCε RA domains have different roles in stability and basal activity. a** Domain structure of rat PLCε. The residue numbers above the diagram correspond to predicted (CDC25-EF1/2) or observed (6PMP, this study) domain boundaries. PLCε variants used in this study are diagrammed below. The open box in PLCε PH-COOH ΔRA1 corresponds to deletion of the RA1 domain. **b** Representative thermal denaturation curves of PH-COOH (black circles), PH-COOH ΔRA1 (blue inverted triangles), PH-RA1 (light green triangles), PH-C2 (purple diamonds), and EF3-RA1 (dark green squares). The most dramatic decreases in thermal stability are in variants that lack the RA1 domain. **c** Loss of both RA domains (PH-C2) or the RA1 domain (PH-COOH ΔRA1) decreases basal-specific activity up to ~20-fold relative to PH-COOH, whereas deletion of RA2 (PH-RA1) only decreased activity ~2-fold. Each data point shown represents the average of the duplicates from one technical repeat. Error bars reflect SD. Data for PLCε PH-COOH was previously reported[22]. Significance was determined using a one-way ANOVA followed by Dunnett's multiple comparisons test vs. PLCε PH-COOH. (****$p \leq 0.0001$, ***$p \leq 0.0005$, **$p \leq 0.001$, *$p \leq 0.05$).

**Table 1 Melting temperature ($T_m$) and basal activity of PLCε variants.**

| PLCε variant | $T_m \pm$ SD (°C) | Significance | $n$ | Specific activity $\pm$ S.D. (nmol IP$_3$/min/ nmol PLCε variant) | Significance | $n$ |
|---|---|---|---|---|---|---|
| PH-COOH[a] | 51.9 ± 0.9 | | 5 | 360 ± 120 | | 9 |
| PH-COOH Δ RA1 | 47.6 ± 1.0 | **** | 3 | 16 ± 4 | **** | 5 |
| PH-RA1 | 50.8 ± 0.3 | | 4 | 200 ± 50 | * | 9 |
| PH-C2[a] | 48.3 ± 1.0 | **** | 4 | 78 ± 20 | *** | 5 |
| EF3-RA1 | 52.7 ± 0.4 | | 4 | 230 ± 30 | | 3 |
| 1526–1546 | 51.0 ± 0.2 | | 3 | 590 ± 40 | ** | 4 |
| 1621–1634 | 50.8 ± 0.3 | | 3 | 290 ± 130 | | 3 |
| N1316E | 48.8 ± 0.4 | **** | 4 | 240 ± 120 | | 3 |
| D1911A | 49.7 ± 0.4 | **** | 4 | 630 ± 120 | *** | 4 |
| R1965A | 50.3 ± 0.6 | ** | 3 | 640 ± 60 | * | 2 |
| F1982A | 50.2 ± 0.3 | *** | 4 | 390 ± 40 | | 3 |
| F1982E | 50.2 ± 0.4 | ** | 3 | 990 ± 300 | **** | 3 |
| F2006A | 48.9 ± 0.3 | **** | 3 | 330 ± 100 | | 3 |
| F2006E | 46.6 ± 0.7 | **** | 4 | 320 ± 50 | | 3 |
| F2077A | 50.4 ± 0.4 | ** | 4 | 400 ± 120 | | 3 |
| R2085A | 49.4 ± 0.2 | **** | 3 | 410 ± 70 | | 3 |

[a]Data previously reported[22].
Results are based on one-way ANOVA followed by Dunnett's multiple comparisons test vs. PLCε PH-COOH. (****$p \le 0.0001$, ***$p \le 0.0005$, **$p \le 0.001$, *$p \le 0.05$).

and in cells. The structure of the PLCε EF3-RA1 fragment, together with functional biochemical and cell-based assays, provides an important step forward in understanding the mechanisms underlying the regulation and architecture of this critical enzyme.

## Results

**The RA1 domain promotes the stability and basal activity of PLCε.** We previously established that fragments beginning at the predicted PH domain (*R. norvegicus* residue 837) could be expressed and purified[22]. Starting from this point, an additional series of catalytically active PLCε domain deletion variants were purified from baculovirus-infected insect cells (Fig. 1a): PH-COOH, the largest PLCε fragment expressed and purified to date, PH-COOH ΔRA1 (internal deletion of RA1 residues 1989–2097); PH-RA1 (truncated at residue 2098); PH-C2 (truncated at residue 1972), and EF3-RA1, which was designed based on the structure of PLCδ[23].

The melting temperatures ($T_m$) of these variants were determined using differential scanning fluorimetry (DSF) (Fig. 1 and Table 1). Variants showing the greatest decrease in $T_m$ relative to PH-COOH were those lacking the RA1 domain: PH-COOH ΔRA1 with a $T_m$ of 47.6 ± 1.0 °C, and PH-C2, with a $T_m$ of 48.3 ± 1.0 °C, both representing ~4 °C decreases (Table 1). The basal-specific activity of these variants was also measured using a liposome-based activity assay[22,24]. The most notable losses of activity were again observed in the PH-COOH ΔRA1 and PH-C2 variants (78 ± 20 and 16 ± 4 nmol IP$_3$/min/nmol PLCε variant, respectively), which are ~22- and ~5-fold lower than PH-COOH (Fig. 1b, c, Table 1). These results demonstrate that the RA1 domain, and the region connecting the C2 and RA1 domains, contribute to the structure and catalytic competency of the PLCε catalytic core. The larger decrease in activity observed for PH-COOH ΔRA1 relative to PH-C2 may also be due in part to an unproductive conformation of the RA2 domain with respect to the PLCε active site (Fig. 2b, c).

**Crystal structure of PLCε EF3-RA1.** The EF3-RA1 variant exhibited a similar $T_m$ and catalytic activity to PH-COOH (Fig. 1 and Table 1), indicating that the PH domain and EF1/2 were dispensable for stability and activity in vitro. This variant also

expressed at ~5–8-fold greater levels than PH-COOH. Thus, to understand how the RA1 domain is integrated with the PLCε core, we determined the 2.7 Å crystal structure of PLCε EF3-RA1 (residues 1284–2192, Fig. 2 and Table 2). There are four copies of PLCε EF3-RA1 in the asymmetric unit arranged as two nearly identical dimers, although there is no evidence of dimerization by size exclusion chromatography or small-angle X-ray scattering (Supplementary Figs. 1 and 2; Supplementary Tables 1 and 2). The dimer interfaces are formed in part by conserved hydrophobic surfaces of EF hands 3/4 (EF3/4) and the TIM barrel domain that might be expected to interact with the PH domain and EF1/2 if they were present, based on structures of other PLC enzymes[12,23,25,26].

In each chain of PLCε, the first helix of the EF3 subdomain (E3α, residues 1284–1303, Supplementary Figs. 1 and 3) is disordered, potentially due to the absence of EF1/2. The PLC common core domains, which span the F3α helix of EF3 through the C2 domain, have a similar arrangement as observed in the PLCβ, PLCδ, and PLCγ subfamilies (Supplementary Fig. 4)[23,25–27]. The highly conserved active site residues within the PLCε TIM barrel are similarly configured to those of other PLC subfamilies, and weak electron density is observed for the catalytic Ca$^{2+}$ in each of the four chains (Supplementary Figs. 1 and 4).

However, the TIM barrel-C2 domain linker and the structure following the C2 domain are dramatically different in PLCε, as obligated by the presence and packing of the RA1 domain (Supplementary Fig. 7). In the EF3-RA1 structure, the C2 and RA1 domains are connected by a 16 residue linker (residues 1973–1990). Although residues 1969–1974 are disordered, the rest of the linker forms two short helices that interact with the TIM barrel and C2 domain, burying ~800 Å$^2$ of surface area. The sidechain of F1982 packs in a hydrophobic pocket formed by residues M1831, F1835, L1842, and M1845 from the loop connecting the TIM barrel and C2 domains, and F1909 from the C2 domain (Fig. 2b). This interface is further stabilized by an electrostatic interaction between R1987 and D1911 from the C2 domain (Fig. 2b, c). The RA1 domain (residues 1991–2093) interacts primarily with EF3/4 through a surface that includes the F3α-E4α loop, and also contacts the C2 domain (Fig. 2d). Some key interactions are made by F2006, which forms hydrophobic interactions with L1308 in EF3/4; F2077, which forms a cation-π interaction and hydrophobic interactions with R1965 and M1967,

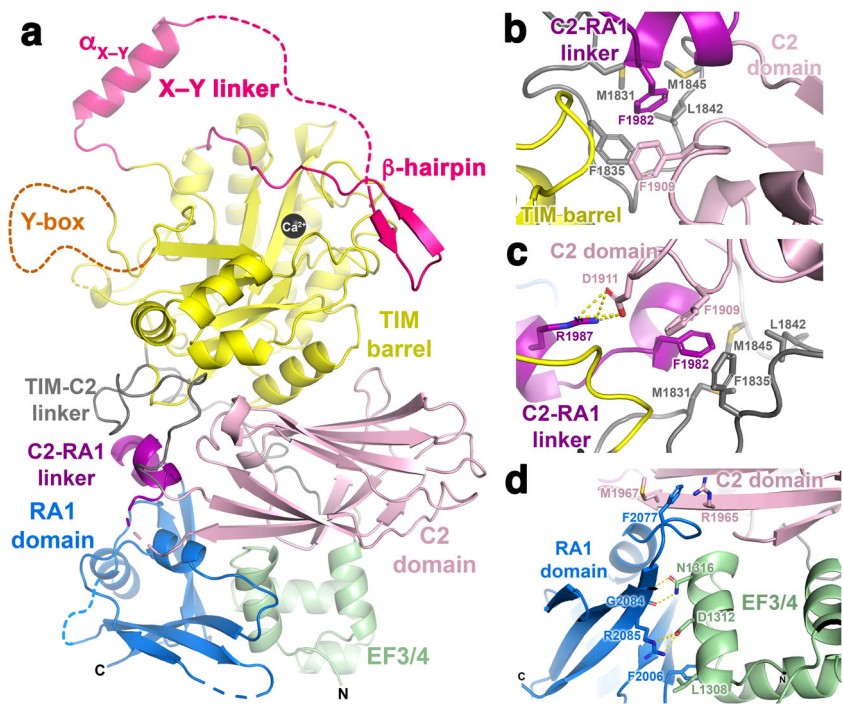

**Fig. 2 The structure of PLCε EF3-RA1 reveals an intimately associated C2-RA1 linker and RA1 domain. a** Crystal structure of PLCε EF3-RA1 with domains colored as in Fig. 1a. The catalytic $Ca^{2+}$ is shown as a black sphere. The overall structure of the RA1 domain in the context of the crystal structure is similar to its solution structure (r.m.s.d. 1.4 Å for 73 Cα atoms, PDB ID 2BYE[18]), with the greatest differences in the loop regions. Dashed lines correspond to disordered loops, and the N- and C-termini of the protein are labeled N and C, respectively. **b** F1982 in the C2-RA1 linker packs in a hydrophobic pocket formed by residues in the TIM barrel-C2 linker (gray) and the C2 domain. **c** R1987 further stabilizes interactions between the C2-RA1 linker and the C2 domain via a salt bridge with D1911. Dashed yellow lines correspond to hydrogen bonds or salt bridges ≤3.5 Å. **d** The RA1 domain interacts with both the C2 domain and the F3α helix of EF3/4. The C-terminus of the RA1 domain is labeled C.

**Table 2 Data collection and refinement statistics (molecular replacement).**

| | PLCε EF3-RA1[a] |
|---|---|
| Data collection | |
| Space group | $P2_1$ |
| Cell dimensions | |
| $a, b, c$ (Å) | 93.6, 127.8, 139.3 |
| $\alpha, \beta, \gamma$ (°) | 90.0 101.1, 90.0 |
| Resolution (Å) | 136.7–2.73 (2.88–2.73)[b] |
| $R_{merge}$ | 0.268 (0.327) |
| $I / \sigma I$ | 5.1 (1.1) |
| Completeness (%) | 99.6 (99.7) |
| Redundancy | 3.4 (3.5) |
| Refinement | |
| Resolution (Å) | 29.9–2.73 |
| No. of reflections | 80,469 |
| $R_{work} / R_{free}$ | 0.234/0.273 (0.327/0.381) |
| No. of atoms | 19,693 |
| Protein | 19,622 |
| Ligand/ion | 4 |
| Water | 67 |
| B-factors | 58.3 |
| Protein | 63.5 |
| Ligand/ion | 109 |
| Water | 38.7 |
| R.m.s. deviations | |
| Bond lengths (Å) | 0.011 |
| Bond angles (°) | 1.7 |

[a]Data was collected from one crystal at GM/CA 23-ID-D.
[b]Values in parentheses are for highest-resolution shell.

respectively; and R2085 and G2084, which form hydrogen bonds with N1316 and D1312 in EF3/4 (Fig. 2d). The RA1 domain buries ~1400 Å² of surface area on the PLCε surface, of which ~600 Å² is through its interaction with EF3, ~400 Å² with the C2 domain, and ~400 Å² with the C2-RA1 linker (Fig. 1).

Like PLCβ and PLCδ, PLCε is autoinhibited by its X–Y linker[12]. This element is poorly conserved across subfamilies with the exception of a 10–15 amino acid acidic stretch proposed to contribute to interfacial activation[12,28–30]. However, the mechanisms of autoregulation, which are required to keep basal PLC activity low, are expected to be unique to each subfamily. In contrast to structures of PLCβ and PLCδ, substantial portions of the PLCε X–Y linker are observed (Fig. 2a and Supplementary Fig. 5). An amphipathic helix (residues 1529–1541, hereafter referred to as the $\alpha_{X-Y}$ helix), is found in all four chains of the asymmetric unit, and a β hairpin (residues 1621–1631) is also observed in one chain (Fig. 2a and Supplementary Figs. 1, 5, and 6). The hydrophobic face of the $\alpha_{X-Y}$ helix packs against EF3/4 of the same chain in an adjacent unit cell of the crystal, in the site that was expected to be occupied by the first helix of EF3, which is disordered in this structure (Supplementary Fig. 6). This interaction seems to be a crystallization artifact, as PLCε EF3-RA1 is monomeric in solution (Supplementary Fig. 2 and Supplementary Tables 1 and 2), and the loop connecting the TIM barrel and a $\alpha_{X-Y}$ helix is too short to allow an intramolecular interaction to form. However, similar intermolecular crystal contacts have been observed for flexible helical regulatory elements in PLCβ that have later proven to be of regulatory importance[25,31]. In chain C, this helix is followed by 78 disordered residues, including the acidic stretch, followed by a β hairpin that packs on two conserved loops on the TIM barrel (Supplementary Fig. 5). This is likely a crystallization artifact,

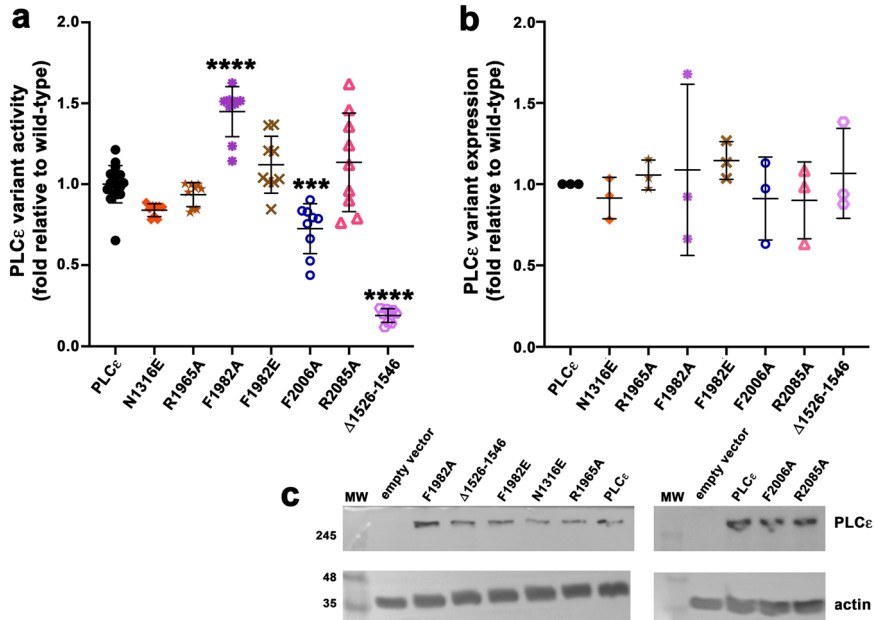

**Fig. 3 Deletion of the $\alpha_{X-Y}$ helix or mutation of the C2-RA1 linker alters lipase activity in cells.** COS-7 cells, which lack endogenous PLCε, were metabolically labeled with [$^3$H]-myoinositol, transfected with PLCε variants, and the amount of [$^3$H]-IP$_x$ quantified by scintillation counting. **a** Deletion of the $\alpha_{X-Y}$ helix decreases lipase activity ~5-fold relative to PLCε. The F2006A mutant, which helps stabilize the RA1-EF3/4 interface, had ~1.4-fold lower activity. In contrast, the F1982A mutant, which disrupts the C2-RA1 linker, increased activity ~1.5-fold. **b** Quantification of western blot data, showing that the differences in activity are not due to changes in expression. **c** Representative western blots, where empty vector and actin were used as the negative control and loading control, respectively. Each data point represents an individual experiment, and error bars reflect SD. Significance was determined using a one-way ANOVA followed by Dunnett's multiple comparisons test vs. PLCε (****$p \leq 0.0001$, ***$p \leq 0.0005$, **$p \leq 0.001$, *$p \leq 0.05$).

given that it is only observed in one chain and is stabilized by crystal packing interactions. However, the interaction is notable as it could help stabilize a twelve residue loop (residues 1631–1643) that passes over the active site, which would need to be displaced for substrate binding (Fig. 2a and Supplementary Fig. 5)[12,28]. In addition to the X–Y linker, the PLCε TIM barrel also contains the Y-box insertion (residues 1660–1729) required for RhoA-dependent activation of PLCε[32,33]. While the majority of the Y-box is disordered in the structure, the loop would be positioned in close proximity to the X–Y linker such that it could readily influence activity via interactions with the linker or the active site (Fig. 2a)[32,33].

**The $\alpha_{X-Y}$ helix contributes to basal activity in a context-dependent manner.** Deletion of the X–Y linker in PLCε increases basal activity ~20-fold in cells[12]. However, the mechanism by which the linker autoinhibits activity is not known. We hypothesized that protein–protein interactions mediated by the $\alpha_{X-Y}$ helix and, potentially the β-hairpin may be involved. To test this idea, we purified two internal deletions in the background of the PH-COOH variant: Δ1526–1546, which removes the $\alpha_{X-Y}$ helix, and Δ1621–1634, which removes the β-hairpin. As these elements make inter- and intramolecular interactions in the crystal structure, respectively, we tested whether their deletion altered thermal stability or basal activity (Table 1 and Supplementary Fig. 8). Both PH-COOH Δ1526–1546 and PH-COOH Δ1621–1634 had $T_m$ values comparable to that of PH-COOH (51.0 ± 0.2 °C and 50.8 ± 0.3 °C, respectively). Whereas PH-COOH Δ1621–1634 had basal activity similar to that of PH-COOH (Table 1), deletion of the $\alpha_{X-Y}$ helix in PH-COOH Δ1526–1546 modestly increased basal activity ~1.6-fold (590 ± 40 nmol IP$_3$/min/nmol PLCε variant), suggesting that in vitro these elements may only play a modest role in autoinhibition in the context of a liposome-based assay.

As the proteins we can purify lack the first 836 amino acids, which could affect the function of these elements, we tested the contribution of the $\alpha_{X-Y}$ helix in the background of full-length PLCε in cells using a [$^3$H]-inositol phosphate (IP$_x$) accumulation assay[34]. Interestingly, the basal activity of PLCε Δ1526–1546 decreased ~5-fold relative to PLCε in this assay (Fig. 3a). This result is not due to decreased expression (Fig. 3b), but likely reflects the more complex environment within the cell and/or loss of interactions between the $\alpha_{X-Y}$ helix and first 836 residues of the PLCε N-terminus that were removed in the PH-COOH variant.

**Interactions of the C2-RA1 linker with the catalytic core inhibit PLC activity.** The crystal structure revealed that the C2-RA1 linker forms extensive interactions with the TIM barrel and C2 domains of the PLCε core (Fig. 2). As deletion of the linker along with the RA1 domain decreases activity and stability (Fig. 1 and Table 1), we hypothesized that perturbation of these interfaces would have a similar effect. We therefore used site-directed mutagenesis to introduce single residue substitutions in the background of PH-COOH. The F1982A and F1982E substitutions were expected to disrupt the interactions of this residue with the hydrophobic pocket formed by the TIM barrel-C2 loop, TIM barrel, and C2 domain (Fig. 2b, c). Indeed, these mutations decreased thermal stability by ~2 °C relative to PH-COOH (50.2 ± 0.3 °C and 50.2 ± 0.4 °C, respectively, Table 1). However, F1982A did not alter basal activity (390 ± 40 nmol IP$_3$/min/nmol PLCε variant), while F1982E had a ~3-fold increase in activity to 990 ± 300 nmol IP$_3$/min/nmol PLCε variant (Table 1 and Supplementary Fig. 8). To further assess F1982A and F1982E in the background of full-length PLCε, their activities were measured in the cell-based [$^3$H]-IP$_x$ assay. The activity of the F1982A mutant was ~1.5-fold higher than basal in this assay, and F1982E also showed a small but insignificant increase in basal activity. These

differences are not due to changes in expression (Fig. 3b, c and Supplementary Fig. 9). Differences in the magnitude of the increase are likely due to the 836 residues at the N-terminus of full-length PLCε, which could contribute to activity through intramolecular interactions with the PLCε core and/or RA domains. We also introduced the D1911A mutation to eliminate the electrostatic interaction with R1987 in the C2-RA1 linker (Fig. 2c). The D1911A mutant decreased thermal stability by ~3 °C relative to PH-COOH ($T_m$ of 49.7 ± 0.4 °C, Table 1). Similar to the F1982E substitution, D1911A also had a ~2-fold increase in activity to 630 ± 120 nmol IP$_3$/min/nmol PLCε variant (Table 1 and Supplementary Fig. 8). Thus, contrary to our expectations, perturbation of the C2-RA1 linker interface in general leads to a modest increase in activity in PH-COOH and full-length PLCε, as measured in two different assay formats at multiple interactions sites.

**Interactions of the RA1 domain with the catalytic core primarily decrease PLCε stability.** To disrupt the RA1–C2 interface, we created R1965A and F2077A in the background of the PH-COOH variant (Fig. 2d). Both mutations decreased the thermal stability by ~2 °C with respect to PH-COOH. The F2077A mutation had no impact on basal activity (Table 1 and Supplementary Fig. 8). However, while the R1965A mutation exhibited a ~2-fold increase in basal activity in vitro (Table 1 and Supplementary Fig. 8), this was not replicated in the background of full-length PLCε in the cell-based assay (Fig. 3 and Supplementary Fig. 9). The reason is unclear, but it could reflect the comparatively small number of contacts between the RA1 and C2 domains, which may be further stabilized through intramolecular contacts mediated by the N-terminal residues present in full-length PLCε relative to PH-COOH.

The interface between RA1 and the F3α helix includes two hydrogen bonds between the G2084 backbone and the sidechain of N1316, and a salt bridge between the sidechains of R2085 and D1312. The interface also features hydrophobic contacts between F2006 and L1308 (Fig. 2d). The R2085A, N1316E, F2006A, and F2006E mutations caused the greatest decreases in thermal stability, with $T_m$ values 3–4 °C lower than that of PH-COOH (Table 1). However, none of these substitutions appreciably altered basal activity in the background of PH-COOH (Table 1 and Supplementary Fig. 8), and only PLCε F2006A showed a slight decrease in basal activity in the cell-based activity assay (Fig. 3). From this data, it seems that although the presence of RA1 and its interactions are generally stabilizing to PLCε, individual point mutants do not have a dramatic effect on basal activity. However, elimination of all RA1 interactions clearly leads to a substantial loss of overall stability (Fig. 1 and Table 1) and coincident loss of basal activity.

## Discussion
PLCε is a critical regulator of calcium and DAG-dependent signaling in the cardiovascular system, where changes in its expression and/or aberrant activation result in cardiac hypertrophy and heart failure[7]. As an integrator of signals from diverse signal transduction pathways, it contains regulatory elements distinct from those of other PLC enzymes, including a unique X–Y linker, a N-terminal CDC25 domain, and C-terminal RA1 and RA2 domains. However, little is known about the molecular mechanisms by which these elements control its basal or G-protein-stimulated activity, due in part to the lack of high-resolution structural information for fragments of PLCε that retain catalytic activity. Our previous functional studies have demonstrated that the PLCε PH domain is dispensable for enzyme stability[22], and our work, along with others, has shown

that the PH, RA domains, and EF1/2 are not required for basal activity[22,32,33]. The RA1 and RA2 domains are known to interact with scaffolding proteins and activated small GTPases, respectively[17,19,20,35], and they have been reported to autoinhibit basal activity. However, how and whether both domains are involved in these processes was unknown[18]. By deleting either RA1 or RA2 in the background of PLCε PH-COOH, we found that RA2 did not contribute to stability and its deletion had only a small effect on basal activity. In contrast, deletion of RA1 decreased stability and basal activity ~20-fold (Fig. 1 and Table 1). Thus, our studies, taken as a whole, indicate that the RA domains themselves do not play a major autoinhibitory role, and, therefore, that their interactions with Ras or Rap GTPases are unlikely to lead to activation via release of autoinhibition.

Although our crystal structure of the PLCε EF3-RA1 fragment shares the conserved core architecture observed in other characterized PLC subfamilies[12,23,25,26], it has been expanded upon by the presence of unique elements in the X–Y linker, the C2-RA1 domain linker, and the RA1 domain, the latter two of which form extensive intramolecular interactions with the canonical PLCε core domains (Figs. 2 and 3). These features are not anticipated to alter the location of the PH domain and EF1/2 with respect to the rest of the enzyme, as compared to the structures of PLCβ and PLCγ[12,23,25,26] (Supplementary Figs. 4 and 6). In fact, conserved patches that instead mediate crystal contacts in the PLCε structure suggest that interdomain interactions between the PH domain, EF hands and TIM barrel will still take place in the intact enzyme (Supplementary Fig. 6). However, these interactions are expected to be transient given that the $T_m$ is similar between PH-COOH and EF3-RA1 (Table 1). We previously showed that deletion of the PH domain did not alter stability but did alter the solution architecture of the variant[22]. Evidence for conformational flexibility of these regions has likewise been reported in PLCδ and PLCβ. In PLCδ, the PH domain binds PIP$_2$ and increases processivity, but is dispensable for lipase activity[36–38]. In PLCβ, the solution structure of the enzyme reveals a more elongated conformation as compared to its crystal structure[22,39], and activation by the Gβγ heterodimer can be blocked by restricting the motion of the PH domain and EF1/2[40]. Our structure of PLCε also provides a structure-based explanation as to why the RA1 domain does not interact with activated GTPases. Superimposing the RA2 domain in the H-Ras–RA2 structure (PDB ID 2C5L[18]) with the RA1 domain in the EF3-RA1 structure shows that the predicted G protein binding surface is occluded by EF3/4 (Supplementary Fig. 10).

The α$_{X–Y}$ helix likely represents a regulatory element that modulates PLCε activity. We found that deletion of the α$_{X–Y}$ helix alters basal activity depending on the assay and the PLCε background. PLCε PH-COOH Δ1526–1546 showed a modest ~1.6-fold increase in basal activity as compared to PH-COOH in the liposome-based assay, whereas the activity of PLCε Δ1526–1546 decreased by ~5-fold in the cell-based assay, nearly to background levels (Table 1 and Fig. 3; Supplementary Fig. 8). Although these results may be partially due to differences in the assay format, we believe the more likely explanation is the additional domain(s) present in the full-length PLCε. The PH-COOH variant lacks the N-terminal 836 amino acids, a highly conserved region that is essentially uncharacterized, with the exception of the CDC25 domain[14,15,21]. The α$_{X–Y}$ helix may interact with the N-terminus of PLCε to modulate activity and/or interactions of the enzyme with the cytoplasmic leaflets of the plasma or perinuclear membranes, either directly or indirectly. Future studies that address the functions of the PLCε N-terminus in basal regulation and that examine the subcellular localization of PLCε as a function of its proposed regulatory elements, such as α$_{X–Y}$, are essential for answering these questions.

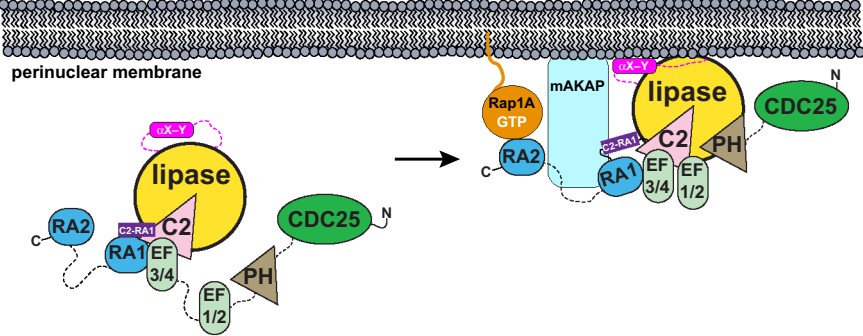

**Fig. 4 Proposed model of basal PLCε regulation at the perinuclear membrane.** (Left) PLCε is present predominantly in the cytoplasm, and is maintained in a low-activity state by the autoinhibitory X–Y linker (hot pink) and the C2-RA1 linker (purple). The CDC25, PH, and RA2 domains, along with EF1/2, are not essential for full basal activity and may be flexibly connected to, or only transiently interact with, the PLCε core[18,22,32,33]. (Right) Localization of PLCε to the perinuclear membrane through interactions between the RA1 domain and the scaffolding protein mAKAP increase lipase activity[17,19]. RA1 binding to mAKAP could alter the conformation of, or displace, the C2-RA1 linker, increasing basal activity. Membrane association would also increase basal activity via interfacial activation[12], which may be facilitated by interactions between the $\alpha_{X-Y}$ helix and the membrane or, alternatively, with other domains in PLCε or proteins at the target membrane, such as activated Rap1A.

The numerous interactions between the C2-RA1 linker and the RA1 domain with the PLCε core provide a structural explanation for the decreased stability and activity of the PH-COOH ΔRA1 and PH-C2 variants (Figs. 1 and 3 and Table 1). Mutation of individual residues in the RA1 interface with the F3α helix of EF3/4 decreased stability, but did not alter basal activity (Fig. 2b–d, Table 1, and Supplementary Fig. 8). Surprisingly, although mutations in the C2-RA1 linker or in the RA1–C2 domain interface (Fig. 2) decreased stability, they generally increased activity across multiple assay formats and construct backgrounds. These results strongly indicate that the C2-RA1 linker region itself may be a functional regulatory element in PLCε, and explain why the PH-COOH ΔRA1 variant has lower activity than that of PH-C2 (Fig. 1).

It is further possible that proteins that bind to RA1 may alter the conformation of the adjacent C2-RA1 linker in a manner that enhances activity. In the cardiovascular system, PLCε is localized to the perinuclear region via interactions between the RA1 domain and the mAKAP scaffolding protein[17,19], where various agonists stimulate PI$_4$P hydrolysis, leading to the accumulation of DAG. In light of our study, it is possible that the increased PLCε activity may also be due to release of autoinhibition by the C2-RA1 linker, potentially mediated through interactions between the RA1 domain and mAKAP, or that prime it for G protein-dependent activation. (Fig. 4). To distinguish between these possibilities, the mAKAP–PLCε interface must be experimentally identified. Autoinhibition of PLC activity via a helix C-terminal to the C2 domain is not unique to PLCε. In the PLCβ subfamily, the Hα2′ helix in its proximal C-terminal domain binds a cleft between the TIM barrel and C2 domains and inhibits basal activity, albeit on the opposite side of the C2 domain (Supplementary Fig. 11). This autoinhibition is released upon interactions with Gα$_q$[31,39], which engages the TIM-C2 loop region of PLCβ3, in a manner more directly analogous to the PLCε C2-RA1 linker. Thus, the TIM barrel-C2 interface of PLCs may be a key regulatory hot spot that has been exploited by nature in different ways among the various PLC isozymes.

## Methods

**PLCε cloning, expression, and purification.** Complementary DNAs (cDNAs) encoding N-terminally His-tagged *R. norvegicus* PLCε variants were subcloned into pFastBac HTA (PH-COOH, residues 837–2282; PH-C2, 832–1972; PH-COOH ΔRA1, 837–2282 Δ1989–2097; PH-RA1, 837–2098; and EF3-RA1, 1284–2098). Point mutants and internal deletions were generated using QuikChange Site-Directed Mutagenesis (Stratagene) or the Q5-site-directed mutagenesis kit (New England BioLabs Inc). All variants were sequenced over the entire coding region. For proteins used in vitro assays and crystallization trials, Sf9 cells were infected with baculovirus encoding PLCε variants at a MOI of 1.0, and harvested after 40–48 h. Cell pellets were flash frozen in liquid nitrogen and stored at −80 °C until purification.

Cell pellets were resuspended in 300 mL lysis buffer containing 50 mM Tris-Cl pH 7.5, 50 mM NaCl, 10 mM β-mercaptoethanol (β-ME), 0.1 mM EDTA, 0.1 mM EGTA, and EDTA-free protease tablets at one-third strength (Roche), and lysed on ice by dounce homogenization. The lysate was clarified by ultracentrifugation at $100{,}000 \times g$ for 1 h. The supernatant was filtered, and applied to GE HisTraps equilibrated with ten column volumes (CV) of buffer A (50 mM Tris-Cl pH 7.5, 300 mM NaCl, 20 mM imidazole, 10 mM β-ME, 0.1 mM EDTA, and 0.1 mM EGTA), then washed with buffer A. The protein was eluted with a gradient of 0–100% buffer A containing 500 mM imidazole. Fractions containing protein were concentrated to 1 mL and applied to a MonoQ column (MonoQ 5/50 GL, GE Life Sciences) equilibrated with Buffer E (50 mM Tris-Cl pH 7.5, 50 mM NaCl, 2 mM DTT, 0.1 mM EDTA, and 0.1 mM EGTA). The protein was eluted with a salt gradient of 0–100% Buffer E supplemented with 500 mM NaCl. Fractions containing protein were pooled, concentrated to 1 mL, and applied to two tandem Superdex 200 10/300 GL columns (GE Healthcare) equilibrated with S200 buffer (50 mM Tris-Cl pH 7.5, 200 mM NaCl, 2 mM DTT, 0.1 mM EDTA, and 0.1 mM EGTA). Fractions corresponding to the purified protein were identified by sodium dodecyl sulfate–polyacrylamide gel electrophoresis (SDS-PAGE), pooled, and concentrated to ~2–3 mg/mL for use in stability and activity assays. The protein was flash frozen in liquid nitrogen, and stored at −80 °C.

PLCε EF3-RA1 used for crystallization was purified as above, with one modification. After elution from the Ni-NTA column, EF3-RA1 was dialyzed with 4% (w/w) TEV overnight at 4 °C against 1.5–2 L of Buffer A. The dialysate was then applied to a Roche cOmplete Ni-NTA column equilibrated with Buffer A, and the flow-through containing the cleaved EF3-RA1 was collected, and passed over the column two more times. The TEV-cleaved EF3-RA1 was subject to further purification as described.

**Crystallization of PLCε EF3-RA1.** PLCε EF3-RA1 was mixed with an equimolar concentration of CaCl$_2$·2H$_2$O. Crystals were obtained with a 1:1 ratio of 6.4 mg/mL PLCε EF3-RA1 mixed with well solution containing 6.25% PEG 4000, 100 mM MES pH 6.00, and 0.2 M NaCl. A slurry of these crystals was then used to seed drops containing a 1:1 ratio 5 mg/mL PLCε EF3-RA1 and well solution containing 100 mM MES pH 5.8, 0.2 M NaCl, and 6.5% (w/v) PEG 8000 at 12 °C.

**Structure determination.** PLCε EF3-RA1 crystals were harvested in a cryoprotectant solution containing 50 mM Tris-Cl pH 7.5, 2.4 M NaCl, 2 mM DTT, 0.1 mM EDTA, 0.1 mM EGTA, 0.1 mM CaCl$_2$-2H$_2$O, 0.2 M MES pH 5.8, and 29% PEG 8000 and flash frozen in liquid nitrogen. Diffraction data was collected at the Advanced Photon Source on the GM/CA 23ID-D beamline using a Pilatus3 6M detector. All data was collected at 110 K at a wavelength of 1.03 Å. Diffraction data was indexed, integrated, and scaled using autoPROC[41]. PLCε EF3-RA1 crystals diffracted to 1.96 Å spacings, but data was truncated to 2.7 Å due to radiation damage and anisotropic diffraction. Initial phases were determined by molecular replacement, using the structure of PLCβ3 (domains EF3-C2, 46% identity, PDB ID 3OHM, Supplementary Figs. 3 and 4[39]) as a search model. The initial model of the

RA1 domain was generated by fitting the NMR structure of RA1 in the electron density (PDB ID 2BYE[18]). The final structural model was generated using alternating rounds of manual building in COOT[42] and restrained refinement with TLS (wherein each individual domain was refined independently) in Refmac[43,44]. Stereochemical correctness was assessed using Molprobity[45] and Procheck[46], with 89.1% of residues in favored regions of the Ramachandran plot and 10.9% in the allowed region. The estimated coordinate error was 0.42 Å. The structure is deposited in the Protein Data Bank under the accession ID of 6PMP.

**Differential scanning fluorimetry**. Melting temperatures ($T_m$) of PLCε variants were determined as previously described[22]. A final concentration of 0.5 mg/mL was used for each PLCε variant. Fluorescence due to protein denaturation was measured as a function of increasing temperature, and the data fit to a Boltzmann sigmoidal function, where the inflection point is used to calculate $T_m$.

**PIP$_2$ hydrolysis assay**. Briefly, 200 μM phosphatidylethanolamine (PE, Avanti), 50 μM PIP$_2$ (Avanti), and ~4000 c.p.m. [$^3$H]-PIP$_2$ (Perkin Elmer) were mixed, dried under nitrogen, and resuspended in sonication buffer containing 50 mM HEPES pH 7, 80 mM KCl, 2 mM EGTA, and 1 mM DTT for each experiment. PLCε variant activity was measured at 30 °C in 50 mM HEPES pH 7, 80 mM KCl, 15 mM NaCl, 0.83 mM MgCl$_2$, 3 mM DTT, 1 mg/mL bovine serum albumin (BSA), 2.5 mM EGTA, 0.2 mM EDTA, and ~500 nm free Ca$^{2+}$. PLCε PH-COOH was assayed at final concentrations of 0.05 ng/μL, 0.075 ng/μL, and 0.1 ng/μL, but approached saturation at 0.1 ng/μL and therefore was not included in the final analysis. PLCε PH-C2 was assayed at a final concentration of 0.1 ng/μL, 0.5 ng/μL, and 1 ng/μL[22], and PH-RA1 at 0.75 ng/μL, 0.1 ng/μL, and 0.5 ng/μL, EF3-RA1 at 0.075 ng/μL, PH-COOH ΔRA1 at 2 or 5μg/μL, and PH-RA1 at 0.5 ng/μL. All PH-COOH point mutants and X–Y linker deletion variants were assayed at a final concentration of 0.1 ng/μL. Control reactions contained everything except free Ca$^{2+}$. Reactions were quenched by the addition of 200 μL 10 mg/mL BSA and 200 μL 10% (w/v) ice-cold trichlororacetic acid and centrifuged for 10 min. 200 μL of the supernatant, which contained [$^3$H]-IP$_3$, was quantified by scintillation counting. Each experiment was performed in duplicate, with each individual experiment performed at least two times on distinct samples. Significance was determined using a one-way ANOVA in which the average specific activity of each variant was compared to that of PH-COOH.

**PLCε cloning and transfection in COS-7 cells**. A pCMVscript vector encoding C-terminally FLAG tagged R. norvegicus PLCε was a gift from A.V. Smrcka. The PLCε point mutations, Δ1526–1546 internal deletion, and empty vector control were generated in this background using the Q5-site-directed mutagenesis kit (New England BioLabs Inc). All plasmids were purified using the Qiagen Hi-Speed Maxi-Tip kit and sequenced over the coding region.

**[$^3$H]-IP$_x$ accumulation assay**. COS-7 cells were plated at a density of 1 x 10$^5$ cells per well in a 12-well plate in high-glucose Dulbecco's Modified Eagle's Medium (Corning) supplemented with 5% fetal bovine serum (FBS, Atlanta Biological), 1X Glutamax (Gibco), 1X penicillin-streptomycin (Corning), at 37 °C with 5% CO$_2$. The following day, cells were transfected with 750 ng/μL DNA encoding PLCε, PLCε variant, or empty vector controls using Fugene-6 (Promega). Approximately 24 h after transfection, cells were washed once with serum-free, inositol-free Ham's F-10 media (Invitrogen), followed by addition of Ham's F-10 media supplemented with 1.5 mCi/well myo[2-$^3$H(N)] inositol (Perkin Elmer) for 16 h. In all, 10 mM lithium chloride was then added to the cells and incubated for 1 h to inhibit the activity of inositol phosphatases. The media was aspirated, and cells were washed once with ice-cold phosphate-buffered saline (PBS), followed by the addition of ice-cold 50 mM formic acid for extraction of [$^3$H]-inositol phosphates. The 12-well plate was incubated on ice for 30 min, after which the solution in each well was transferred to Dowex AGX8 anion exchange columns to isolate the inositol phosphates. Columns were washed with once with 50 mM formic acid, once with 100 mM formic acid, and then the inositol phosphates were eluted with buffer containing 1.2 M ammonium formate and 0.1 M formic acid into scintillation vials and counted. All experiments were performed at least three times in triplicate.

**Western blotting**. Cells were lysed in RIPA buffer (150 mM sodium chloride, 1% octylphenoxy poly(ethyleneoxy)ethanol (IGEPAL-CA 630, Sigma Aldrich), 0.5% sodium deoxycholate, 0.1% SDS, and 25 mM Tris pH 7.4) and protease inhibitor tablets at 1X strength (Roche). Whole-cell lysates were incubated on ice for 15 min, then centrifuged for 15 min at 13,300 x g at 4 °C, and the supernatant removed for immunoblotting. The cleared lysate was mixed with loading buffer, and all samples were incubated at 90 °C for 10 min before loading onto a 6% SDS-PAGE gel. The proteins were transferred to a polyvinylidene fluoride membrane overnight. The following day, the membrane was blocked with 5% BSA dissolved in 1X TBST for 1 h, followed by incubation with the primary antibodies: an anti-FLAG mouse antibody (1:1000) and anti-actin mouse antibody (1:5000) (Cell Signaling Technology) overnight. The membrane was then watched three times with 1X TBST, and incubated with the goat anti-mouse secondary antibody conjugated with HRP

(Sigma Aldrich) for 1 h. The membrane was washed three times with 1X PBS, followed by addition of the ECL substrate (Thermo Fisher) and imaging. Band intensity was quantified by densitometry using the Image J software[47]. All experiments were performed at least three times.

**Statistics and reproducibility**. All assays using purified protein were performed at least three times as technical duplicates or triplicates, using protein from separate preparations as much as possible. Significance was determined using one-way ANOVA, followed by Dunnett's multiple comparisons test in which the average measurement for each variant was compared to PH-COOH. In the [$^3$H]-IP$_x$ accumulation assay, experiments were performed at least three times using technical triplicates, and significance was determined using one-way ANOVA, followed by Dunnet's multiple comparisons test, in which the average measurement for each variant was compared to full-length PLCε.

**Sequence alignments**. The amino acid sequences of R. norvegicus PLCε (UNIPROT Q99P84), R. norvegicus PLCδ (UNIPROT P10688), H. sapiens PLCβ2 (UNIPROT Q00722), and H. sapiens PLCβ3 (UNIPROT Q01970) were aligned using Clustal-Omega[48] and Jalview[49]. For regions with lower sequence conservation (>40% identity or similarity), the sequences were manually aligned and confirmed by superposition of the crystal structures of PLCε EF3-RA1 (PDB ID 6PMP), PLCδ (PDB ID 2ISD[23]), PLCβ2 (PDB ID 2ZKM[12]), and PLCβ3 (PDB ID 3OHM[39]).

**Small-angle X-ray scattering (SAXS) data collection and analysis**. PLCε EF3-RA1 was diluted to final concentrations of 2–3 mg/mL in S200 buffer and centrifuged at 16,000 x g for 5 min at 4 °C prior to data collection. Size exclusion chromatography (SEC)-SAXS was performed at the BioCAT beamline at Sector 18 of the Advanced Photon Source (Supplementary Table 1).

EF3-RA1 was eluted from a Superdex 200 Increase 10/300 GL column using an ÄKTA Pure FPLC (GE Healthcare) at a flow rate of 0.7 mL/min (Supplementary Fig. 2a. The eluate passed through a UV monitor and through a 1.5 mm ID quartz capillary with 10 μm walls for data collection. Scattering intensity was recorded with a Pilatus3 X 1M detector (Dectris) placed ~3.7 m from the sample using 12 KeV X-rays (1.033 Å wavelength) and a beam size of 160 x 75 μm, allowing for an accessible $q$ range of ~0.004 Å$^{-1}$ to 0.36 Å$^{-1}$. Data was collected every 2 s with 1 s exposure times. Data in the regions flanking the elution peak was averaged, creating buffer blank peaks, which were then subtracted from the elution peak exposures to generate the final scattering profile (Supplementary Fig. 2A)[50]. BioXTAS RAW 1.4.0[50] was used for data processing and analysis. The radius of gyration ($R_g$) of individual frames were plotted with the scattering chromatograms, which plot integrated intensity of individual exposures as a function of frame number, and used to help determine appropriate sample ranges for subtraction. PRIMUS[51] was used to calculate the $R_g$, $I_{(0)}$, and $D_{max}$ for the samples. Graphical plots were generated from buffer-subtracted averaged data (scattering profile and Guinier plots[52]) and plotted using GraphPad Prism v.8.0.1. SAXS data are presented in accordance with the publication guidelines for small-angle scattering data[53].

**Reporting summary**. Further information on research design is available in the Nature Research Reporting Summary linked to this article.

## Data availability
The structure factors and coordinates for PLCε EF3-RA1 are deposited in the Protein Data Bank under accession ID 6PMP. All source data underlying the figures presented in the main text is available in the Dryad general depository[54].

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

## Acknowledgements

We thank I. J. Fisher, R. Gil de Rubio, and A. V. Smrcka (U. Michigan) for DNA encoding *R. norvegicus* PLCε, COS-7 cells, and helpful discussions. This work is supported by an American Heart Association Predoctoral Fellowship Grant 18PRE33990057 (M.S.), American Heart Association Scientist Development Grant 16SDG29920017 (A.M.L.), an American Cancer Society Institutional Research Grant (IRG-14-190-56) to the Purdue University Center for Cancer Research (A.M.L.), and NIH 1R01HL141076-01 (A.M.L.). Use of the Advanced Photon Source, an Office of Science User Facility operated for the U. S. Department of Energy (DOE) Office of Science by Argonne National Laboratory, was supported by the U.S. DOE under Contract Number DE-AC02-06CH11357. The content is solely the responsibility of the authors and does not necessarily represent the official views of the National Heart, Lung, and Blood Institute or the National Institutes of Health.

## Author contributions

N.Y.R., E.E.G.-K., M.S., K.M., and A.M.L. designed the experimental approach. N.Y.R., E.E.G.-K., M.S., K.M., H.O., W.M., A.F.S., A.T.M., A.E., E.M., and A.M.L. cloned, expressed, and purified PLCε variants. M.S., E.E.G.-K., K.M., M.M.V.C., A.F.S., A.T.M., A.E., E.M., and A.M.L. performed DSF and activity assays. N.Y.R. crystallized and solved the structure of PLCε EF3-RA1. N.Y.R., E.E.G.-K., M.S., K.M., and A.M.L. wrote the manuscript.

## Competing interests

The authors declare no competing interests.
