## [Peer Review File · Communications Biology]

Reviewers' comments:

Reviewer #1 (Remarks to the Author):

In 'Crystal structure of phospholipase C ϵ reveals structural elements important for activity in cells and regulation through its C-terminal RA domains', Rugema and colleagues present the crystal structure of a fragment of the protein Phospholipase C ϵ (PLC ϵ), an enzyme involved in the regulation of the cardiovascular homeostasis through the generation of lipids second messengers. Particularly, the authors investigate the role of the Ras association (RA) domains in the enzyme activity, as well as the role of a newly discovered regulatory structural element of the enzyme in the X-Y linker domain. The main conclusion of this work is that an amphipathic helix in the X-Y linker, namely α X-Y, is able to control the activity of the enzyme both in vitro and in cells.

My overall opinion on the manuscript is that is very well written and structured, with nicely organised figures and clear description of the results. However, I found that some results were contradictory and might benefit of additional investigation or clarification. I also identified few very minor issues in each section of the manuscript, that will be found listed below.

Introduction.

The introduction was very well written and organised, clear, and concise. I would suggest that a reference to Figure 1A is provided after the description of the domain organisation.

Methods.

Since Communications Biology does not have any word limit, I would suggest that the authors present the Supplementary Methods as part of the main manuscript (therefore removing them from the Supplementary Information – SI).

PLC ϵ cloning, expression, and purification.

- 1) There is a typo in line 5: sequenced instead of "sequence".
- 2) Another typo at line 10: the word 'in' is missing before "300 mL...".
- 3) Some of the information presented in the third paragraph could be omitted/summarised since already present in the second paragraph of the same section (for example: the buffers of the MonoQ purifications are the same as in the second paragraph, so something like 'same buffer as above were used' could be stated).

PIP2 hydrolysis assay.

- 1) The method reports that the "reactions were quenched by the addition of 200 mL"; it seems like there is a typo and mL should be substituted with μ L.
- 2) The authors state that "All assays were performed at least twice in duplicate". I interpreted this as that each experiment was performed in duplicates, but the experimental repeats were more than 2 (as reported in Table 1); it might be worth clarifying this.

Western blotting.

- 1) the antibodies concentration is missing.

Results.

The RA1 domain promotes the stability and basal activity of PLC ϵ .

- 1) Figure 1B's caption reports that "Each data point represents an experiment performed in duplicate". Could the authors clarify whether this means that each data point in the plot is the average of the duplicates of each technical repeat?
- 2) Table 1 reports the significance by using a system of letters reported as superscript; such system is quite unusual and I think it would be easier for the reader to visualise the significance using the more common "asterisks method" (which is by the way used in the rest of the figures and manuscript). I would then add columns next to the ones reporting Tm and Specific activity with the asterisks

corresponding to the p values described in the caption.

3) Table 1 reports specific activity values for each of the construct analysed, measured in nmolProd/min/nmolenz. The parameter used to measure the enzymatic activity is correct. The specific activity is measured as the slope of a straight line fitting the amount of product (or substrate depleted) over time with increasing enzyme amounts. However, according to the PIP2 hydrolysis assay section, a range of enzyme concentration was tested only for the PH-C2 (0.01-1 ng/μL), while for all the other enzymes it looks like only one concentration was tested: can the authors clarify how the specific activity was calculated? If only one enzyme concentration was tested, this could generate great error in the measurements and the experiments should be repeated (see R. Eisinger, M.J. Danson, "Enzyme assays," Second, Oxford University Press, 2002).

Crystal structure of PLCε EF3-RA1.

1) The authors state that "no evidence of dimerization by size exclusion chromatography or small angle X-ray scattering" was observed. Data should be reported in Supplementary Information.

2) It is reported that "In each chain of PLCε, the first helix of the EF3 subdomain is disordered". How was this assessed? Was the B-factor calculated from the electron density map? If so, this should be reported, or stated otherwise that it was not possible to fit this helix in the density map.

3) Any explanation for the fact that only 3X Ca²⁺ were found in the crystallographic tetramer? Is the tetramer assembling in the space unit possibly affecting the access to Ca²⁺ to one of the monomers? What is the rmsd of the alignment of the "apo" monomer with the Ca²⁺-bound monomer? Answers to these questions should be included in the caption of Figure S1.

4) The RCSB PDB report on the crystal structure deposition highlighted few potential issues with the quality of the fitting. The residue-property plots indicate a great number of outliers in the sequence of all the monomers. Moreover, a long list of too-close contacts was identified. Can the authors comment on this? Do they think this is due to the relatively low homology between PLCε and the proteins used for the molecular replacement (PLCβ3 PDB ID 3OHM – 46% identity; RA1 domain PDB ID 2BYE)? If so, was MR tested with other members of the PLC family? How many cycles of refinements were performed? Is any of the outlier involved in the interactions described later in the manuscript?

5) This results section refers to Supplementary Figure 2, which reports sequence alignments between various members of the protein family in different species. As the figures stands, it is very difficult to appreciate the conservation between paralogs. I would correct it by introducing a colour system to highlight the conservation. In my experience, I found the software JalView very useful to generate figures that are visually easy to interpret, but I am sure there are few more software that could help with this. Same applies to Supplementary Figure 4.

Note also that the authors should report how the alignments were performed in the methods.

6) In Figure 2, the TIM-C2 linker (grey) is not indicated as all the other structural elements.

7) The authors describe the inter-molecular interaction between the αX-Y of one of the monomers and the EF3/4 of the same chain in an adjacent unit cell of the crystal. This region would be expected to be occupied by the first helix of EF3, which is disordered in this structure. However, the authors state that "The loop between the TIM barrel and the αX-Y helix is too short to allow formation of the same interaction in cis". At the same time, it was also stated that "there is no evidence of dimerization by size exclusion chromatography or small-angle X-ray scattering". Considering all the reported statements, I would be inclined to believe that the interaction between αX-Y and the first helix of EF3 of the adjacent unit cell is perhaps a crystallographic artifact. Can the authors comment on this? Did the authors explore this interaction into more details? The manuscript refers to previous SAXS data: could these data be revised to identify evidence of movement of the αX-Y observed to indicate an interaction between the two structural elements? Is there any indication of interaction in solution between the αX-Y and the EF3 (by ITC, or MST, or NMR, or SPR, or BLI, or FRET)? This should be investigated and reported in the manuscript, also to corroborate some of the results presented in the following section.

The αX-Y helix modulates basal regulation in a context-dependent manner.

In this section, the effects of the mutations of amino acids of the X-Y linker on the activity is analysed.

Some of the results generated in vitro and in vivo did not agree and I think a deeper investigation should be attempted to explain some of the results obtained.

1) The authors "hypothesised that the α X-Y helix and/or β -hairpin [of X-Y linker] may be involved" in a regulation of the activity. However, in the previous section they also reported that "the loop between the TIM barrel and the α X-Y helix is too short to allow formation of the same interaction in cis". The activity of the mutant PH-COOH Δ 1526-1546 showed significantly increased activity in vitro. This would corroborate a role of this α X-Y helix, but also stresses the importance of understanding whether this happens in cis, or in trans in a putative dimer (see previous comments).

2) When the Δ 1526-1546 deletion was applied to the full-length protein in vivo, a decrease of the activity was observed, in disagreement with the in vitro studies. The authors state that this "supports the idea that this element has a profound effect on regulating PLC ϵ activity". In my opinion this is not supporting a regulatory role, but instead, I am afraid, is confusing. This is not followed up by additional explanations in the Discussion. How do the authors explain this? Could the mutation in the context of the full-length induce (partial) unfolding of the protein, therefore affect the activity? Interactions of the C2-RA1 linker with the catalytic core autoinhibit PLC activity.

1) Similarly to what observed for the Δ 1526-1546 mutant, no agreement between in vitro and in vivo results was observed for other mutants presented in this section (F1982E, F1982A, R1965A), but no actual discussion is provided. How is this explained?

2) Why was the D1911A not studied in in vivo assays?

Overall, it seems that the in vivo assays disagree with the in vitro assays and the X-ray crystallography analysis. My inclination is that the in vivo assays require better optimisation and/or repeats.

Discussion.

1) I noticed a typo in the sentence "activation by the G β γ heterodimer can be blocked by restriction the motion of the PH domain and EF1/2", the preposition 'of' is missing.

2) It would be probably better to define the Figure 5 caption as 'proposed' model, since it is based on speculations only.

Overall, I recommend the publication of this paper upon the suggested amendments and additional analysis/experiments, since that the scientific community would benefit of the new structure reported for drug discovery purposes.

Moreover, I wish the authors and their families health and some semblance of stability in these trying times.

Kind regards and stay safe.

Reviewer #2 (Remarks to the Author):

Rugema et al provide detailed molecular insights into the structural architecture of phospholipase C epsilon, providing a crystal structure of the minimal catalytically active core, consisting of two EF hands, TIM barrel, C2, and RA1 domains. Importantly, the structure explains why PLC ϵ is not activated by G proteins compared to PLC β and adds clarity to regulatory mechanisms. Overall, this is a careful and well performed study that will significantly add to the understanding of PLC epsilon function. Reorganization of the results and inclusion of a structural model will greatly help the general reader.

Specific comments:

The impact of the structure and overall manuscript will be greatly enhanced with discussion of how the entire molecule may come together at higher resolution than Fig 5. Starting with Fig 2A, where is the PH domain thought to bind? Supplemental Fig 3 is referenced with respect to PH and EF1/2 and the other PLC's (top of pg 14) but there is no indication of where these structures lie in this figure. The authors also speculate that "hydrophobic patches that mediate crystal contacts" actually form interdomain interactions. What domains? And where are these conserved hydrophobic patches? These thoughts should be better crystallized in an overall model.

1. The organization of the results section should be revised. For example, the authors don't mention panels 2B or 2C until after extensive discussion of Figure 3. It would make more sense (as currently written) to make 2B and 2C a separate figure after Fig 3. Moreover, it is not clear why the authors include figure 2B outside of supplemental when it is just a crystal packing element/artifact. Does alphaX-Y:EF3/4 contacts have any physiological or functional relevance? For clarity, authors should state that Fig 2C is the same chain, 180 degree view from 2A.
2. Bottom of pg 7, the authors state that the "RA1 domain itself interacts primarily with the C2 domain", yet this interaction is not shown in any detail (don't need the word "itself"). Figures detail interactions with EF3/4. The authors should state exactly the amount of buried surface area of RA1-C2 versus RA1-EF3/4. This is also true for the bottom of pg 8 for surface area interactions with the RA1 domain – each interaction with PLC core domain (which one?) should be detailed to add up to ~1700 A.
3. The authors state that the RA2 domain has only a minor role, yet deletion of both RA domains is far less deleterious to activity than deletion of just RA1. Is this due to the large deletion and thus the wrong orientation of RA2 or due to deletion of the C2-RA1 linker as well?
4. Top of page 10, second line – not clear why Supplementary Figure 4 is referenced?
5. Pg 11, Fig 4: The authors assume that lipase activity assays in COS-7 cells measure only basal activity, yet is it not possible that other cellular regulators account for at least some of the measured activity. Does the relative pattern of activity change with growth factor stimulation? Have the authors confirmed similar localization for WT, delta alphaX-Y, and F1982A?
6. Bottom of pg 11, please indicate specific PLCe core domain that interacts with F1982 and F1982.
7. A number of awkward sentences throughout. For example 2nd sentence on pg 4 of results (remove "which" for each construct).
8. Statistical analysis and methods are appropriate.

We thank the Editor and Reviewers for their helpful feedback. We have responded to the comments in line below. All changes described below have been highlighted in light gray in the revised manuscript.

Reviewer #1:

The introduction was very well written and organised, clear, and concise. I would suggest that a reference to Figure 1A is provided after the description of the domain organisation.

Thank you, and we added a call to Figure 1A after the description of the PLC ϵ domain organization.

Methods.

Since Communications Biology does not have any word limit, I would suggest that the authors present the Supplementary Methods as part of the main manuscript (therefore removing them from the Supplementary Information – SI).

We have consolidated the method sections and have moved the information that was formerly in the supplement into the main text.

In response to the suggestion that we include the size exclusion chromatogram and the small angle X-ray scattering (SAXS) data which confirms the PLC ϵ EF3-RA1 variant is monomeric in solution, we have updated the supplementary information to include the SAXS methods, data collection and analysis tables (Supplementary Tables 1 and 2), and a figure (Supplementary Figure 2).

PLC ϵ cloning, expression, and purification.

- 1) There is a typo in line 5: sequenced instead of “sequence”.**
- 2) Another typo at line 10: the word ‘in’ is missing before “300 mL...”.**
- 3) Some of the information presented in the third paragraph could be omitted/summarised since already present in the second paragraph of the same section (for example: the buffers of the MonoQ purifications are the same as in the second paragraph, so something like ‘same buffer as above were used’ could be stated).**

Thank you. We have corrected the typos and have condensed the expression and purification protocols.

PIP2 hydrolysis assay.

- 1) The method reports that the “reactions were quenched by the addition of 200 mL”; it seems like there is a typo and mL should be substituted with μ L.**
- 2) The authors state that “All assays were performed at least twice in duplicate”. I interpreted this as that each experiment was performed in duplicates, but the experimental repeats were more than 2 (as reported in Table 1); it might be worth clarifying this.**

We have corrected the typo in the volume. We have also clarified the description of replicates and numbers of experiments. The text now includes the following statement: “Each experiment was performed in duplicate, with each individual experiment performed at least two times on distinct samples.”

Western blotting.

- 1) the antibodies concentration is missing.**

We have updated the methods to include the anti-FLAG mouse antibody was used at a 1:1000 dilution and the anti-actin mouse antibody was used at a 1:5000 dilution.

Results.

The RA1 domain promotes the stability and basal activity of PLC ϵ .

1) Figure 1B's caption reports that "Each data point represents an experiment performed in duplicate". Could the authors clarify whether this means that each data point in the plot is the average of the duplicates of each technical repeat?

In these experiments, each data point represents the average of the duplicates measured in a single technical repeat. We have updated the caption to include this information.

2) Table 1 reports the significance by using a system of letters reported as superscript; such system is quite unusual and I think it would be easier for the reader to visualise the significance using the more common "asterisks method" (which is by the way used in the rest of the figures and manuscript). I would then add columns next to the ones reporting T_m and Specific activity with the asterisks corresponding to the p values described in the caption.

We have revised the table to include two additional columns reporting significance by the suggested asterisks method for both the T_m and basal specific activities for the PLC ϵ variants.

3) Table 1 reports specific activity values for each of the construct analysed, measured in nmolProd/min/nmolenz. The parameter used to measure the enzymatic activity is correct. The specific activity is measured as the slope of a straight line fitting the amount of product (or substrate depleted) over time with increasing enzyme amounts. However, according to the PIP₂ hydrolysis assay section, a range of enzyme concentration was tested only for the PH-C2 (0.01-1 ng/ μ L), while for all the other enzymes it looks like only one concentration was tested: can the authors clarify how the specific activity was calculated? If only one enzyme concentration was tested, this could generate great error in the measurements and the experiments should be repeated (see R. Eisinger, M.J. Danson, "Enzyme assays," Second, Oxford University Press, 2002).

For PH-C2, we measured the specific activity by quantifying the amount of [³H]-IP₃ at five time points and fitting the increase in product as a function of time. The slope of the resulting straight line was then divided by the concentration of protein used in each experiment (0.1, 0.5, and 1 ng/ μ L). However, the [³H]-PIP₂ substrate became commercially unavailable over the course of this study. Given that this assay is the gold standard in the field, we did not want to change assays midstream, and so we accepted that we would not be able to repeat the same procedure for every variant. This was the biggest challenge for PH-C variants that were generated latest in the study. Thus in the end, the PH-COOH, PH-C2, and PH-RA1 variants were tested at three different concentrations over a five time point curve. PH-C Δ RA1 was tested at two concentrations over a five time point curve. The PH-COOH site directed mutants were however tested at one concentration with a five time point curve. We are comfortable with the PH-COOH mutants at one protein concentration because they were tested over five time points, and we have thoroughly optimized the assay with wild-type PH-COOH at 0.1 ng/ μ L, which allows for a good basis of comparison. Therefore, we do not feel additional assays are necessary, and in any event, there is no more available [³H]-PIP₂. For these reasons, we hope you can agree with us.

Our approach to the assays is potentially being questioned given some inconsistency in the results obtained with purified proteins using this assay, and what is observed in cells. We performed the cell-based assays to enhance the rigor of our approach as a check on our *in vitro* results, in part because it is not yet possible to purify full-length PLC ϵ . Overall, the results obtained were not dramatically different with the exception of the α_{X-Y} deletion mutant, which hopefully should reassure readers (and we

emphasize this point in the revision). We talk more about the α_{X-Y} to address another point below, but in a nutshell, we accept the fact that some variants are going to behave differently in our cell-based assay than they do in a test tube in a liposome-based assay. In fact, such mutants often turn out to be the most interesting because they imply regulatory mechanisms not yet reproduced with purified components. In our minds, the 5-fold decrease in activity observed in cells is probably more significant than the modest 1.5 fold increase measured *in vitro*, if only because we use full-length PLC ϵ in cells.

Crystal structure of PLC ϵ EF3-RA1.

1) The authors state that “no evidence of dimerization by size exclusion chromatography or small angle X-ray scattering” was observed. Data should be reported in Supplementary Information.

We have now added a new Supplementary Figure 2, which includes (A) the elution profile of PLC ϵ EF3-RA1 from size exclusion columns, the (B) SAXS raw scattering curve, (C) Guinier plot, and (D) pair-distance distribution function (P(r) plot) We have also two supporting tables for the SAXS-derived parameters of EF3-RA1 (Supplementary Table 1) and for the data collection and analysis (Supplementary Table 2) as established by Trewhella and coworkers (Trewhella, J. *et al*, *Acta Cryst D* **73**(9), 710-728 (2017)).

2) It is reported that “In each chain of PLC ϵ , the first helix of the EF3 subdomain is disordered”. How was this assessed? Was the B-factor calculated from the electron density map? If so, this should be reported, or stated otherwise that it was not possible to fit this helix in the density map.

The PLC ϵ EF3-RA1 construct spans residues 1284-2098, but there was no interpretable electron density observed for residues 1284-1302 in any of the chains. Therefore, this region was not modeled and there is no way to calculate a B factor.

In its expected position, we observed strong helical density that was ultimately identified as the α_{X-Y} helix through iterative model building and refinement, with clear electron density connecting it to the TIM barrel of an adjacent chain in each of the four chains of PLC ϵ in the asymmetric unit. In our improved model, the average B-factors for residues 1524-1544 were 73.8 Å² (chain A), 79.4 Å² (chain B), 86.0 Å² (chain C), and 82.6 Å² (chain D.) The overall average B-factor for the structure was 58.3 Å².

3) Any explanation for the fact that only 3X Ca²⁺ were found in the crystallographic tetramer? Is the tetramer assembling in the space unit possibly affecting the access to Ca²⁺ to one of the monomers? What is the rmsd of the alignment of the “apo” monomer with the Ca²⁺-bound monomer? Answers to these questions should be included in the caption of Figure S1.

The density observed for Ca²⁺ in the PLC ϵ EF3-RA1 active sites is weak, as compared to the Ca²⁺ density observed in crystal structures PLC β and PLC δ . We have revised the model building and refinement procedures to improve the structure and address concerns raised regarding the PDB validation report. F_o-F_c difference Fourier maps generated peaks consistent with Ca²⁺ in each of the four chains. This information is now included in the caption of Supplemental Figure 1.

4) The RCSB PDB report on the crystal structure deposition highlighted few potential issues with the quality of the fitting. The residue-property plots indicate a great number of outliers in the sequence of all the monomers. Moreover, a long list of too-close contacts was identified.

Can the authors comment on this? Do they think this is due to the relatively low homology between PLC ϵ and the proteins used for the molecular replacement (PLC β 3 PDB ID 3OHH – 46% identity; RA1 domain PDB ID 2BYE)? If so, was MR tested with other members of the PLC family? How many cycles of refinements were performed? Is any of the outlier involved in the interactions described later in the manuscript?

We understand the basis for your concerns and have taken several steps to address issues related to the modeling of PLC ϵ EF3-RA1. Although it was evidenced by the anisotropic overall B factors, we failed to mention that the diffraction data for these crystals were anisotropic, meaning that the resolution of the structure is not 2.7 Å in all directions (2.7 Å along h, 3.0 Å along k, and 3.5 Å along l). Thus, the fit of the data is not going to be comparable to structures with isotropic 2.7 Å diffraction. There are also many loops in this structure that are quite mobile and on their own would be difficult to model; but by considering the density for each of the four models in the asymmetric unit, using NCS restraints, and relying on precedence from other PLC structures, we can build these loops with reasonable certainty. However, the density is not such that it will be able to avoid minor clashes after unbiased refinement. Hence the clash list and the RSZR outliers may seem large in number (also consider the fact that we do have four copies of everything in the ASU, so multiply all clashes by ~4).

The use of PLC β 3 EF3-C2 domains as our search model for molecular replacement is highly unlikely to be responsible for these issues, as we also used the PLC δ EF3-C2 structure (PDB ID 2ISD) as a search model in parallel, which gave a nearly identical hit.

In our initial submission, we carried out over 70 rounds of iterative model building and refinement prior to depositing the structure in the PDB. To ensure that we explored all avenues for improving our structure, we sought out the expertise of an outside consultant with over 30 years of experience in X-ray crystallography. They personally went through our data, including analysis, model building, and refinement strategies, and found no major cause for concern. Nonetheless, we were able to improve the model in significant ways by fixing a few stereochemical errors, typically in rotamer choice where density was ambiguous, and by applying a more sophisticated TLS model that allowed individual domains to refine their parameters independently. All Ramachandran outliers listed were thoroughly investigated and are consistent with the electron density, or are found in tight turns that justify these kinds of outliers (implying distortion of the amide bond to accommodate, but our resolution will not allow us to model this). This additional ~10 rounds of refinement reduced the total number of close contacts to a total of 161, corresponding to 40-43 in each chain. As each molecule in the asymmetric unit has ~800 residues, these too-close contacts represent less than 5% of the residues, comparable to that observed in other structures of similar resolution as shown in the PDB Validation Report. The vast majority of these close contacts involve hydrogen atoms. Others, such as the contact between 1809 O and 1814 HA2 observed in each chain, are well supported by the electron density and prior PLC structures, and may be an example of a rare carbon-oxygen H-bond. However, at our resolution we would not want to comment on it.

Of special concern are the residues we identified as being key to intramolecular contacts in the structure. Of the seven amino acids subjected to site-directed mutagenesis (N1316, D1911, R1965, F1982, F2006, F2077, R2085), only F2077 was involved in a close contact with a hydrogen (in Met1967).

5) This results section refers to Supplementary Figure 2, which reports sequence alignments between various members of the protein family in different species. As the figures stand, it is very difficult to appreciate the conservation between paralogs. I would correct it by introducing a colour system to highlight the conservation. In my experience, I found the software JalView

very useful to generate figures that are visually easy to interpret, but I am sure there are few more software that could help with this. Same applies to Supplementary Figure 4.

Note also that the authors should report how the alignments were performed in the methods.

We have updated the sequence alignments shown in Supplementary Figures 3 and 5 to include color coding to reflect sequence conservation as identified using both Jalview and Clustal-Omega. Conserved residues are highlighted in a dark shade, while similar residues are highlighted in a lighter shade. We have also updated the methods section to explain how the sequence alignments were performed.

6) In Figure 2, the TIM-C2 linker (grey) is not indicated as all the other structural elements.

We have updated Figure 2A such that the TIM barrel-C2 linker is included.

7) The authors describe the inter-molecular interaction between the $\alpha X-Y$ of one of the monomers and the EF3/4 of the same chain in an adjacent unit cell of the crystal. This region would be expected to be occupied by the first helix of EF3, which is disordered in this structure. However, the authors state that “The loop between the TIM barrel and the $\alpha X-Y$ helix is too short to allow formation of the same interaction in cis”. At the same time, it was also stated that “there is no evidence of dimerization by size exclusion chromatography or small-angle X-ray scattering”. Considering all the reported statements, I would be inclined to believe that the interaction between $\alpha X-Y$ and the first helix of EF3 of the adjacent unit cell is perhaps a crystallographic artifact. Can the authors comment on this?

We believe the interaction observed between the α_{X-Y} helix and EF3/4 from a related molecule is indeed a crystallization artifact. The interaction likely formed during crystallization in order to bury the hydrophobic face of the α_{X-Y} helix and the hydrophobic surface on EF3 that was exposed due to the F3 α helix being disordered. Thus, we believe this particular interaction would not be expected to form in the full-length enzyme, or in fragments of PLC ϵ retaining the PH domain and EF1/2. The surface of EF3 exposed in our PLC ϵ EF3-RA1 structure participates in interdomain contacts, as observed in the structures of the core domains of PLC β and PLC δ (PDB ID 2ZKM, 3OHM, and 2ISD).

We have revised this section of the manuscript in several ways. First, we have moved the figure panel showing the interactions between the α_{X-Y} helix and EF3/4 to Supplementary Figure 5. This figure also highlights the location of the exposed hydrophobic surface of EF3 in the context of the EF3-RA1 structure, how the interface would be buried in the presence of the PH domain and EF1/2, and how the F3 α helix would interact with the hydrophobic surface of EF3. We have also moved the figure panel showing the interactions between the β -hairpin and the TIM barrel to Supplementary Figure 7. Finally, we have revised the text to explicitly state the interaction between α_{X-Y} and EF3 is a crystallization artifact and provided the rationale with supporting experimental data (Supplementary Figure 2, Supplementary Tables 1, 2).

Did the authors explore this interaction into more details? The manuscript refers to previous SAXS data: could these data be revised to identify evidence of movement of the $\alpha X-Y$ observed to indicate an interaction between the two structural elements? Is there any indication of interaction in solution between the $\alpha X-Y$ and the EF3 (by ITC, or MST, or NMR, or SPR, or BLI, or FRET)? This should be investigated and reported in the manuscript, also to corroborate some of the results presented in the following section.

We observed no evidence of dimerization or higher-order oligomerization of PLC ϵ EF3-RA1 by size exclusion chromatography or by small angle X-ray scattering (Supplementary Figure 2, Supplementary Tables 1, 2). If this interaction were to form in solution, one would expect that PLC ϵ EF3-RA1 would polymerize into an extended chain. We see no evidence of this. Given that the interaction represents a crystallization artifact, we did not carry out any further studies to investigate the interactions between α_{X-Y} and EF3-RA1.

The α_{X-Y} helix modulates basal regulation in a context-dependent manner.

In this section, the effects of the mutations of amino acids of the X-Y linker on the activity is analysed. Some of the results generated in vitro and in vivo did not agree and I think a deeper investigation should be attempted to explain some of the results obtained.

1) The authors “hypothesised that the α_{X-Y} helix and/or β -hairpin [of X-Y linker] may be involved” in a regulation of the activity. However, in the previous section they also reported that “the loop between the TIM barrel and the α_{X-Y} helix is too short to allow formation of the same interaction in cis”. The activity of the mutant PH-COOH Δ 1526-1546 showed significantly increased activity in vitro. This would corroborate a role of this α_{X-Y} helix, but also stresses the importance of understanding whether this happens in cis, or in trans in a putative dimer (see previous comments).

We apologize for the confusion about how we interpret the observed interaction between the α_{X-Y} helix and the EF3/4 domain. All PLCs have an X-Y insertion within the TIM barrel domain that is critical for their regulation, and structural and functional studies have shown that the ordered elements within the linkers are often key to this process. For example, in PLC γ , the regulatory insertion within the TIM barrel is comprised of a split PH domain, and two SH2 domains, and one SH3 domain that are required for phosphorylation-dependent activation (Hajicek, N. *et al Elife* **8** e51700 (2019)). In PLC β , crystal structures revealed the C-terminal end of the X-Y linker formed a short helix that interacted with residues adjacent to the active site, acting as a lid to regulate substrate binding. Displacement of this helix was shown to occur via a mechanism involving interfacial activation (Hicks, S.N. *et al Mol Cell* **31**(3), 383-394 (2008), Charpentier, T.H. *et al J. Biol. Chem.* **289**(43), 29545-29557 (2014), Lyon, A.M. *et al Structure* **22**(12), 1844-1854 (2014), Hudson, B.N. *et al Biochemistry* **58**(32), 3454-3467 (2019)). In the present study, we sought only to test whether the α_{X-Y} helix within the X-Y linker might modulate basal activity, with no expectation that it might mediate dimerization or bind to EF3 in cis. We now only try to emphasize the fact that this conserved element has the capacity to a) form a helix and b) form protein-protein interactions. The question now is with what? In other PLCs these types of features have been observed in crystal structures and most have turned out to be functionally relevant.

2) When the Δ 1526-1546 deletion was applied to the full-length protein in vivo, a decrease of the activity was observed, in disagreement with the in vitro studies. The authors state that this “supports the idea that this element has a profound effect on regulating PLC ϵ activity”. In my opinion this is not supporting a regulatory role, but instead, I am afraid, is confusing. This is not followed up by additional explanations in the Discussion. How do the authors explain this? Could the mutation in the context of the full-length induce (partial) unfolding of the protein, therefore affect the activity?

We apologize for the poor explanations for these findings. Our goal in these experiments was solely to test whether the α_{X-Y} helix within the X-Y linker impacted basal activity. As we have historically been a structural biology/biochemistry lab, our general strategy is to work with purified components in clearly defined systems, such as the [3 H]-PIP $_2$ hydrolysis assay. While we would ideally express mutations in the background of full-length proteins and test the purified proteins *in vitro*, this is not yet possible for

PLC ϵ . Instead, we are restricted to the PLC ϵ PH-COOH variant, the largest fragment purified to date, which lacks 836 residues at the N-terminus. With this caveat, we expressed and purified the PH-COOH Δ 1526-1546 variant for our *in vitro* studies and found deletion of α_{X-Y} increased basal activity ~1.6-fold relative to PH-COOH. This is a small difference in basal specific activity in this assay, particularly when compared to the fold change in activity upon deletion of one or more RA domains. This was however satisfying because the X–Y linker is known to impose autoinhibition.

To be rigorous, we also tested our variants in cell-based assays, in particular the [3 H]-IP $_x$ assay in COS-7 cells, but now the background of full-length PLC ϵ . In this assay, the same Δ 1526-1546 deletion decreased activity more dramatically (~5 fold). We do not believe the decrease is due to partial unfolding, or any folding defects, as the protein was expressed at levels comparable to wild-type PLC ϵ .

We believe the difference in the change in basal activity measured for this deletion between the two assays reflects the complex role that the X–Y linker plays in regulating PLCs. It not only serves as a lid over the active site that inhibits activity in solution, but also is intimately involved in interactions with other partners in cells (be it the membrane itself or other regulatory proteins) that either recruit the enzyme to specific locals or trigger displacement of the lid during interfacial activation. Loss of α_{X-Y} in living cells may thus have a profound impact on activity, whereas in an *in vitro* assay, where substrate and enzyme concentrations are far higher than in cells, the most noticeable role of the linker could instead be inhibition of basal activity. The difference is at the very least intriguing, and we believe it does belie a regulatory role for this helix. It will thus serve as a springboard for future studies.

We would also like to point out that it is common for mutations to have dramatically different effects in purified proteins than in cells. For example, mutation of a surface in the Leukemia-associated RhoGEF (LARG) PH domain had no effect in solution, but eliminated activity when the protein was expressed in cells (Aittaleb, M. *et al Cell Signal.* **21**(11), 1569-1578 (2009)). It was later shown the loss of activity was due to the finding that the PH domain is used to bind RhoA·GTP, thus creating a feed-forward loop important for sustaining RhoA signaling and gene expression in cells (Chen, Z. *et al J. Biol. Chem.* **285**(27), 21070-21081 (2010); Dada, O. *et al J. Struct. Biol.* **202**(1), 13-24 (2018)).

Elements of this discussion are now included in the revised manuscript.

Interactions of the C2-RA1 linker with the catalytic core autoinhibit PLC activity.

1) Similarly to what observed for the Δ 1526-1546 mutant, no agreement between *in vitro* and *in vivo* results was observed for other mutants presented in this section (F1982E, F1982A, R1965A), but no actual discussion is provided. How is this explained?

In the first draft we chose to focus discussion on the mutations that had the most dramatic effect. The manuscript was an automatic resubmission through the Nature journal system, and the prior target journal did not allow for an extended discussion. It was clearly our mistake to not expand our discussion to cover all the mutants upon resubmission.

The F1982A/E mutations were introduced to disrupt the interaction between the C2-RA1 linker and the PLC ϵ core domains. The F1982A mutation did not increase basal activity in the background of PH-COOH *in vitro*, but increased activity ~1.5 fold in the background of full-length PLC ϵ and activity measured in the cell-based assay. The F1982E mutation increased activity ~3-fold in the PIP $_2$ hydrolysis assay and a mild but insignificant increase in the cell-based assay. Thus, mutation of this position increases activity in both assay formats, with the magnitude of the increase dependent upon the background of the mutation. In combination with the D1911A mutation, they all are consistent with C2-RA1 linker playing a role in autoinhibiting PLC ϵ activity. The difference in the fold increase found in

each assay has a few explanations, such the fact that full-length PLC ϵ contains 836 residues at the N-terminus that may contribute to activity via intramolecular interactions that are missing in the PH-COOH background and concentration and environment of the substrate PIP₂ is quite different in each context.

The C2-RA1 interface was tested by the R1965A mutation in both the *in vitro* and *in vivo* assays. In the liposome-based assay, the activity of PH-COOH R1965A was ~2-fold higher than PH-COOH. However, in the background of PLC ϵ in the cell-based assay, there was no significant change in activity. This difference is not due to changes in expression, but likely reflects the comparatively small number of contacts between the RA1 and C2 domains, which may be further stabilized by the N-terminal residues present in full-length PLC ϵ . This discussion is now included in the revised manuscript.

2) Why was the D1911A not studied in in vivo assays?

The residues selected for the cell-based assays were chosen based on number of interactions made by the residue and/or the total buried surface area, such that each interface was represented. We therefore decided to follow up on the F1982A/E mutations because they were considered likely to be critical for maintaining the hydrophobic core of the interface. Disruption of D1911 was expected to play a more subtle role because it instead forms a solvent exposed salt bridge.

In summary, we viewed the cell-based assays as a necessary step to enhance the rigor of our analysis, and we believe they were performed well, with robust and reproducible signal to noise, and enough repetitions. We also feel it is acceptable (and also interesting) if the *in vitro* and *in vivo* assay results sometimes differ, so long as they are not for technical reasons (e.g. poor expression or gross incompetence). After all, there are many things yet to learn about the structure of full length PLC ϵ and about its interactions in the complex milieu of the cell that are as of yet difficult to predict from the structure of the protein itself. We hope our expanded discussions of our analyses will satisfy readers on this point.

Discussion.

1) I noticed a typo in the sentence “activation by the G $\beta\gamma$ heterodimer can be blocked by restriction the motion of the PH domain and EF1/2”, the preposition ‘of’ is missing.

Thank you for catching this.

2) It would be probably better to define the Figure 5 caption as ‘proposed’ model, since it is based on speculations only.

We have changed the title for Figure 5 to “Proposed model of basal PLC ϵ regulation at the perinuclear membrane.”

Reviewer #2:

Specific comments:

The impact of the structure and overall manuscript will be greatly enhanced with discussion of how the entire molecule may come together at higher resolution than Fig 5. Starting with Fig 2A, where is the PH domain thought to bind? Supplemental Fig 3 is referenced with respect to PH and EF1/2 and the other PLC’s (top of pg 14) but there is no indication of where these structures lie in this figure. The authors also speculate that “hydrophobic patches that mediate crystal contacts” actually form interdomain interactions. What domains? And where are these

conserved hydrophobic patches? These thoughts should be better crystallized in an overall model.

Thank you for the suggestions. We have added additional supplementary figures to illustrate the expected locations of the PH and EF1/2 domains within PLC ϵ . In Supplementary Figure 5, we have included a structure of PLC ϵ EF3-RA1 that highlights the conserved, exposed hydrophobic and electrostatic residues that would be expected to interact with the PH and EF1/2 domains. Crystal structures of PLC β (PDB ID 2ZKM, 3OHM) and PLC γ (PDB ID 6PBC) reveal consistent positions for the PH domain and EF1/2. To generate a model of where the PH domain and EF1/2 would be located with respect to the rest of the core domains, we superimposed PLC ϵ EF3-C2 on the structure of the PLC β 3 core (PDB ID 3OHM, 46% sequence identity). In the resulting model, the PH domain and EF1/2 are poised to engage the TIM barrel, EF3/4, and C2 domains, with no steric hindrance due to the RA1 domain in PLC ϵ EF3-RA1. We also show that the *in trans* interaction between PLC ϵ EF3/4 and α_{X-Y} observed in the crystal structure recapitulate the expected interaction between the F3 α helix and the EF3 subdomain, based on other PLC structures.

1. The organization of the results section should be revised. For example, the authors don't mention panels 2B or 2C until after extensive discussion of Figure 3. It would make more sense (as currently written) to make 2B and 2C a separate figure after Fig 3. Moreover, it is not clear why the authors include figure 2B outside of supplemental when it is just a crystal packing element/artifact. Does α_{X-Y} :EF3/4 contacts have any physiological or functional relevance? For clarity, authors should state that Fig 2C is the same chain, 180° degree view from 2A.

Thank you for the suggestions. We have made the following changes to Fig. 2 in the revised manuscript. First, we have combined the crystal structure with the detailed interactions of the C2-RA1 linker and RA1 domain with the PLC ϵ core, simplifying the discussion of results, and providing a more concise description of the PLC ϵ specific interfaces. We agree the α_{X-Y} helix-EF3/4 interaction is a crystal packing artifact (although evidence of its potential to form protein-protein interactions), and have moved the figure to the supplementary information, along with details interaction of the β -hairpin with the TIM barrel domain (Supplementary Figures 5, 7). Although we believe the α_{X-Y} helix is a functionally important element that could potentially bind to other proteins or perhaps to the membrane, we regret the original text and figures failed to clearly express this point and resulted in confusion.

2. Bottom of pg 7, the authors state that the “RA1 domain itself interacts primarily with the C2 domain”, yet this interaction is not shown in any detail (don't need the word “itself”). Figures detail interactions with EF3/4. The authors should state exactly the amount of buried surface area of RA1-C2 versus RA1-EF3/4. This is also true for the bottom of pg 8 for surface area interactions with the RA1 domain – each interaction with PLC core domain (which one?) should be detailed to add up to ~1700 Å².

We have updated the text to provide additional information regarding the extent of the interactions formed by the C2-RA1 linker, RA1 domain, and the PLC ϵ core domains. The interactions between the C2-RA1 linker (residues 1973-1990), with the TIM barrel-C2 loop, the TIM barrel, and C2 domains buries ~800 Å² surface area. The RA1 domain buries ~1400 Å² on the PLC ϵ surface, of which ~600 Å² is through its interaction with EF3, ~400 Å² from the C2 domain, and ~400 Å² from the C2-RA1 linker. We have included this information in the main text.

3. The authors state that the RA2 domain has only a minor role, yet deletion of both RA domains is far less deleterious to activity than deletion of just RA1. Is this due to the large deletion and thus the wrong orientation of RA2 or due to deletion of the C2-RA1 linker as well?

We believe the ~22-fold decrease in the basal specific activity of PH-COOH Δ RA1, compared to PH-COOH, is due to retention of the C2-RA1 linker and loss of the RA1 domain. This is in part because in the PH-C2 variant, which lacks both C2-RA1 linker and the RA domains, the activity is only decreased ~5-fold, consistent with the C2-RA1 linker autoinhibiting activity. However, you are correct that we cannot exclude the possibility that an incorrect orientation of the RA2 domain in PH-COOH Δ RA1 may also contribute to the decreased activity measured for this protein. We have updated the text to reflect this additional consideration.

4. Top of page 10, second line – not clear why Supplementary Figure 4 is referenced?

We have removed this erroneous call, thank you for bringing it to our attention.

5. Pg 11, Fig 4: The authors assume that lipase activity assays in COS-7 cells measure only basal activity, yet is it not possible that other cellular regulators account for at least some of the measured activity. Does the relative pattern of activity change with growth factor stimulation? Have the authors confirmed similar localization for WT, delta alpha-y, and F1982A?

We used COS-7 cells for the lipase activity assays as they have very low endogenous expression of PLC enzymes, along with low expression of upstream signaling components, including G α subunits and growth factor receptors (Wu, D. *et al J. Biol. Chem.* **267**(3), 1811-1817 (1992); Woon, C.W. *et al J. Biol. Chem.* **264**(10), 5687-5693 (1989); Osawa, S., *et al Cell* **63**, 697-706 (1990); Cotecchia, S. *et al PNAS* **85**, 7159-7163 (1988); Jiang, H. *et al J. Biol. Chem.* **271**(23), 13430-13434 (1996). In addition, they are also easy to grow and have high transfection efficiency, making them a well-established cell-line for these types of experiments.

Under the assay conditions used in this study, 24 h after transfection with control or PLC ϵ variant, the cells are washed with inositol-free Ham's F-10 media (Invitrogen), then incubated with serum-free and inositol-free Ham's F-10 media supplemented with 1.5 mCi/well myo[2-³H(N)] inositol (Perkin Elmer) for 16 h. Given that the cells are serum-starved for 16 h, it seems unlikely that activation of endogenous growth factor signaling pathways occurs under these conditions. Furthermore, as all cells are subjected to the same washing and labeling steps, and the activity of the PLC ϵ mutants is normalized to the wild-type activity, any differences due to residual growth factors from the media should be minimal. However, we have not directly tested growth-factor stimulation of COS-7 cells transfected with PLC ϵ or variants, as this was beyond the scope of the present study.

In future studies we aim to test differences in localization among our variants and upon activation of different surface receptors. We have only thus far been able to confirm using immunofluorescence that wild-type PLC ϵ is located in the cytoplasm of COS-7 cells, consistent with prior reports (Song, C. *et al J. Biol. Chem.* **276**(4), 2752-2757 (2001); Song, C. *et al Oncogene* **21**(53), 8105-8113 (2002); Smrcka, A.V. *et al Cell Signal* **24**(6), 1333-1343 (2012)).

6. Bottom of pg 11, please indicate specific PLC ϵ core domain that interacts with F1982 and F1982.

We have updated the text to reflect that F1982 interacts with a hydrophobic pocket formed by the TIM barrel-C2 loop, the TIM barrel, and C2 domain.

7. A number of awkward sentences throughout. For example 2nd sentence on pg 4 of results (remove “which” for each construct).

Thank you for the feedback, we have made every effort to improve clarity.

REVIEWERS' COMMENTS:

Reviewer #1 (Remarks to the Author):

I am overall very satisfied by the amendments and the additional data and clarifications provided by Rugema et al regarding the manuscript entitled 'Crystal structure of phospholipase C ϵ reveals structural elements important for activity in cells and regulation through its C-terminal RA domains'.

Amendments on all sections are satisfactory and fully met my requests reported in the previous round of review. I am particularly pleased by the addition of the SEC-SAXS data and the rewriting of the results sections concerning the X-ray crystallography section and the mutants analysis. The revised format of these sections is now significantly clearer. I am also very pleased with the extensive discussion provided for the mutants analysis and I fully agree on the way the data are carefully presented, pointing at some of the limitations of this study, as well as their potential importance for future investigations.

I would also like to thank the authors for the extensive explanations and discussions provided in the rebuttal letter for all the points reported in my previous round of review.

Having said that, I strongly recommend the publications of this manuscript, upon very minor amendments:

1. In the previous round of review, I questioned the authors regarding how the specific activity measurements were performed. The authors have fully clarified the issues and I agree on the experimental procedure used to calculate the enzymatic activity, particularly considering the challenge of dealing with the commercial unavailability of the radiolabelled substrate. There is still something that needs some clarification though. In the revised manuscript, the authors state that "PLC ϵ PH-COOH and EF3-RA1 were assayed at a final concentration of 0.075 ng/ μ L [...] and PH-RA1 at 0.5 ng/ μ L", therefore, only one concentration seems to have been tested for these two constructs. However, in the rebuttal letter, the authors explained that "PHCOOH, PH-C2, and PH-RA1 variants were tested at three different concentrations over a five time point curve". The exact range of concentration should then be reported in the methods for these three constructs.

Moreover, according to the rebuttal letter, the authors state that "the PH-COOH site directed mutants were however tested at one concentration with a five time point curve - 0.1 ng/uL". For statistical significance reasons, the comparison between mutants and WT should then be carried considering only the data obtained with the PH-COOH WT 0.1 ng/uL (as opposed to the average between the specific activity obtained at the three different WT enzyme concentrations tested). I believe that a clarification for this should be reported in the methods.

2. The sentence "However, this interaction is notable as this type of interaction could help stabilize a 12 residue loop (residues 1631-1643) that passes over the active site, which would need to be displaced for substrate binding (Figure 2A, Supplementary Figure 5)" could be perhaps rephrased to avoid the repetition of the word 'interaction' too many times;

3. In the sentence "These differences are not due changes in expression (Figure 3)", there is a typo: the word 'to' is missing.

I would like to thank the authors for their contribution to the PLC field and wish them and their families health and a quick return to normality in these trying times.

Kind regards and stay safe.

Reviewer #2 (Remarks to the Author):

Revisions to the manuscript were thoughtful and comprehensive, addressing all concerns.

We thank the Editor and Reviewers for their helpful feedback. We have responded to the comments in line below. All changes described below have been highlighted in light gray in the revised manuscript.

Reviewer #1:

I would also like to thank the authors for the extensive explanations and discussions provided in the rebuttal letter for all the points reported in my previous round of review.

Having said that, I strongly recommend the publications of this manuscript, upon very minor amendments:

1. In the previous round of review, I questioned the authors regarding how the specific activity measurements were performed. The authors have fully clarified the issues and I agree on the experimental procedure used to calculate the enzymatic activity, particularly considering the challenge of dealing with the commercial unavailability of the radiolabelled substrate. There is still something that needs some clarification though. In the revised manuscript, the authors state that “PLC ϵ PH-COOH and EF3-RA1 were assayed at a final concentration of 0.075 ng/ μ L [...] and PH-RA1 at 0.5 ng/ μ L”, therefore, only one concentration seems to have been tested for these two constructs. However, in the rebuttal letter, the authors explained that “PH-COOH, PH-C2, and PH-RA1 variants were tested at three different concentrations over a five time point curve”. The exact range of concentration should then be reported in the methods for these three constructs. Moreover, according to the rebuttal letter, the authors state that “the PH-COOH site directed mutants were however tested at one concentration with a five time point curve – 0.1 ng/ μ L”. For statistical significance reasons, the comparison between mutants and WT should then be carried considering only the data obtained with the PH-COOH WT 0.1 ng/ μ L (as opposed to the average between the specific activity obtained at the three different WT enzyme concentrations tested). I believe that a clarification for this should be reported in the methods.

Thank you for bringing this to our attention and apologize for the confusion. We measured the basal activity of the PH-COOH, PH-C2, and PH-RA1 variants at three different protein concentrations (PLC ϵ PH-COOH at 0.05 ng/ μ L, 0.075 ng/ μ L, 0.1 ng/ μ L, PH-C2 at 0.1 ng/ μ L, 0.5 ng/ μ L, and 1 ng/ μ L, and PH-RA at 0.75 ng/ μ L, 0.1 ng/ μ L, and 0.5 ng/ μ L). For the PH-COOH variant, the data included in the final version of the manuscript and used in the specific activity comparisons was the average of the specific activity determined using the enzyme at 0.05 ng/ μ L and 0.075 ng/ μ L, because the assay reached saturation at 0.1 ng/ μ L and thus a rate at this concentration could not be accurately determined. All of this information is now provided in the methods section.

Because of this, it is not technically possible for us to compare the specific activity of the PH-COOH variant at the same concentration as the PH-COOH variants. We have now made it clear in the methods how we are doing the comparison. However, we believe it is valid to compare specific activities measured at different enzyme concentrations as they are normalized for the amount of enzyme in the sample, so long as there is no differences in oligomeric state that comes about from changes in concentration (which we have no evidence of); this may be what the reviewer is concerned about in this case. In any event, comparisons of enzyme specific activity measured at different concentrations becomes unavoidable when some variants have exceptionally low or high activity and go out of range of the assay, as is common for comparing mutants in this activity assay. To reassure ourselves and the reviewer, we performed the same statistical comparisons by comparing the specific activities of the PH-COOH variants (measured at 0.1 ng/ μ L) to the specific activity of PH-COOH measured at either 0.05 ng/ μ L or at 0.075 ng/ μ L. In either case, we obtained the same trends in significance using a one-way ANOVA comparing the variants to the averaged specific activity for wild-type PH-COOH.

2. The sentence “However, this interaction is notable as this type of interaction could help stabilize a 12 residue loop (residues 1631-1643) that passes over the active site, which would need to be displaced for substrate binding (Figure 2A, Supplementary Figure 5)” could be perhaps rephrased to avoid the repetition of the word ‘interaction’ too many times.

It now reads “However, the interaction is notable as it could help stabilize a twelve residue loop (residues 1631-1643) that passes over the active site, which would need to be displaced for substrate binding (Figure 2A, Supplementary Figure 5)^{12,28}.”

3. In the sentence “These differences are not due changes in expression (Figure 3)”, there is a typo: the word ‘to’ is missing.

Thank you for bringing this to our attention. We have corrected the sentence.